# Generation of nanobodies from transgenic 'LamaMice' lacking an endogenous immunoglobulin repertoire

Thomas Eden [1,12], Alessa Z. Schaffrath[1,12], Janusz Wesolowski [1,12], Tobias Stähler[1], Natalie Tode[1], Nathalie Richter[1], Waldemar Schäfer [1], Julia Hambach [1], Irm Hermans-Borgmeyer[2], Jannis Woens[3], Camille M. Le Gall [4], Sabrina Wendler[5], Christian Linke-Winnebeck [5], Martina Stobbe[5], Iwona Budnicki[6], Amelie Wanney [6], Yannic Heitz[6], Lena Schimmelpfennig[6], Laura Schweitzer [6], Dennis Zimmer[6], Erik Stahl [7], Fabienne Seyfried[1], Anna J. Gebhardt[1], Lynn Dieckow [1], Kristoffer Riecken [3], Boris Fehse [3], Peter Bannas [8], Tim Magnus [9], Martijn Verdoes [4], Carl G. Figdor [4], Klaus F. Hartlepp [5], Hubertus Schleer[6], Jonas Füner[7], Nicola M. Tomas [10], Friedrich Haag [1], Björn Rissiek [9], Anna M. Mann [1], Stephan Menzel [1,11] & Friedrich Koch-Nolte [1] ✉

Due to their exceptional solubility and stability, nanobodies have emerged as powerful building blocks for research tools and therapeutics. However, their generation in llamas is cumbersome and costly. Here, by inserting an engineered llama immunoglobulin heavy chain (IgH) locus into IgH-deficient mice, we generate a transgenic mouse line, which we refer to as 'LamaMouse'. We demonstrate that LamaMice solely express llama IgH molecules without association to Igκ or λ light chains. Immunization of LamaMice with AAV8, the receptor-binding domain of the SARS-CoV-2 spike protein, IgE, IgG2c, and CLEC9A enabled us to readily select respective target-specific nanobodies using classical hybridoma and phage display technologies, single B cell screening, and direct cloning of the nanobody-repertoire into a mammalian expression vector. Our work shows that the LamaMouse represents a flexible and broadly applicable platform for a facilitated selection of target-specific nanobodies.

The increasing demand for nanobodies in research, biotechnology and medicine[1-7] is attributable to their excellent biochemical properties, including small size and high solubility[8-11]. Nanobodies, the single variable domains (VHHs) of camelid heavy chain antibodies (hcAbs)[12], are ideally suited for imaging and high-resolution microscopy[13-19], can serve as chaperones for crystallization and cryo-electron microscopy[20-24], and provide excellent antigen-binding domains for chimeric antigen receptor (CAR)-T cells and adeno-associated virus

[1]Institute of Immunology, University Medical Center Hamburg-Eppendorf, Hamburg, Germany. [2]Center for Molecular Neurobiology Hamburg, University Medical Center Hamburg-Eppendorf, Hamburg, Germany. [3]Research Department Cell and Gene Therapy, University Medical Center Hamburg-Eppendorf, Hamburg, Germany. [4]Department of Medical BioSciences, Radboud University Medical Center, Nijmegen, The Netherlands. [5]ChromoTek GmbH, Martinsried, Germany - A part of Proteintech Group, Martinsried, Germany. [6]Genovac GmbH, Freiburg, Germany. [7]Preclinics GmbH, Potsdam, Germany. [8]Department of Radiology, University Medical Center Hamburg-Eppendorf, Hamburg, Germany. [9]Department of Neurology, University Medical Center Hamburg-Eppendorf, Hamburg, Germany. [10]III. Department of Medicine, University Medical Center Hamburg-Eppendorf, Hamburg, Germany. [11]Present address: Core Facility Nanobodies, University of Bonn, Bonn, Germany. [12]These authors contributed equally: Thomas Eden, Alessa Z. Schaffrath, Janusz Wesolowski. ✉e-mail: nolte@uke.de

(AAV) gene therapy vectors[25,26]. Their robustness also allows easy genetic fusion to other nanobodies and proteins to generate constructs such as biparatopic and bispecific antibodies[3,27] or tethered G protein-coupled receptor-ligands[28]. Two nanobody-based drugs have entered the clinic: caplacizumab, a bivalent nanobody directed against von Willebrand factor[29], and cilta-cel, a CAR-T cell immunotherapy containing a pair of biparatopic nanobodies against B cell maturation antigen (BCMA/CD269) as antigen-binding domains[30].

Traditionally, nanobodies are derived from immunized camelids or synthetic VHH libraries[9,10,31–35]. Immunization harnesses the clonal expansion and affinity maturation of natural immune responses. The husbandry of camelids, however, is labor-intensive, time-consuming and expensive. Mice offer facile breeding as well as easy access to lymphoid organs and to genetic tools. Several immunoglobulin-transgenic (Ig-tg) mice and rats were designed for the discovery of immunoglobulin single variable domains[36–38], most of which rely on the use of human VH domains. However, human VH and CH1 domains exhibit an inherent tendency to pair with a VL and a CL domain, respectively, via hydrophobic amino acid residues. Consequently, effective induction of hcAb responses in those mice requires deletion of the gene loci for CH1 domains and additional deletion of the gene loci for the κ and λ light chains. Recently, an Ig-tg mouse model ("nanomouse") was reported, in which the VH genes of the mouse IgH locus were replaced by a semisynthetic DNA encompassing camelid VHH genes flanked by mouse regulatory elements[39]. Since the exon encoding CH1, i.e., the first constant domain of the heavy chain that mediates binding to the light chain, was not inactivated for all endogenous mouse IgH-isotypes, nanomice produce a mixture of hcAbs and conventional antibodies.

In this study, we generated a new Ig-tg mouse line that allows the production of llama hcAbs. To this end, we engineered a bacterial artificial chromosome (BAC) encompassing the essential elements of the llama IgH locus and introduced this BAC as a transgene into mice harboring an inactivated endogenous IgH locus. The resulting mice express llama IgM and IgG2b hcAbs without any detectable association with endogenous mouse λ or κ light chains. Using a diverse set of antigens, we demonstrate the straightforward discovery of target-specific nanobodies from LamaMice using classical hybridoma and phage display technologies, single B cell screening, and direct cDNA cloning and sequencing.

## Results

### Cloning and engineering of BACs from the llama IgH locus

From a *Lama glama* BAC genomic DNA library, we isolated two BACs that overlap by ~10 kb and together cover the ~200 kb core region of the llama IgH locus (Fig. 1a). With the exception of a duplicated IgG1 locus, the sequences of BACs V03 and F07 are colinear with cosmid clones from the alpaca IgH locus[40]. Using BAC recombineering, we introduced five modifications into BAC V03: i) to increase the diversity of the VHH-repertoire, we inserted five additional $V_H$H genes, including a

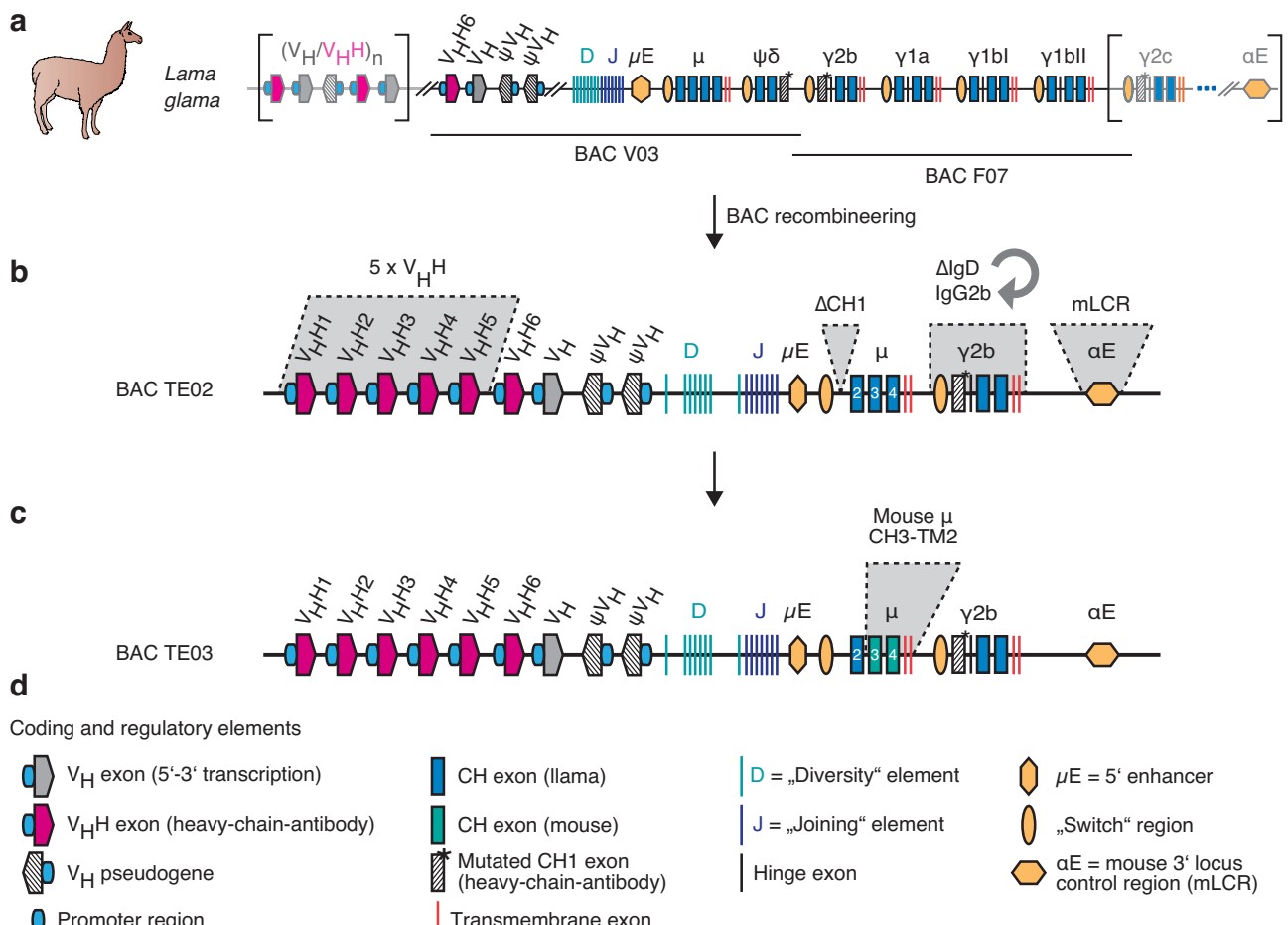

**Fig. 1 | Recombineering of llama IgH BAC transgenes. a** Two bacterial artificial chromosomes (BAC V03, BAC F07) covering the core of the llama IgH locus were cloned from a liver genomic DNA library and fully sequenced. **b** BAC recombineering was used to insert five additional $V_H$H-genes, to delete the CH1 exon of IgM, to replace the IgD pseudogene with the IgG2b gene, and to insert elements of the mouse 3' locus control region (mLCR, αE), resulting in BAC TE02. **c** BAC recombineering was used to exchange the exons encoding the CH3, CH4, transmembrane, and cytosolic domains of IgM with corresponding murine sequences, resulting in BAC TE03. **d** Explanation of symbols used in the schematics.

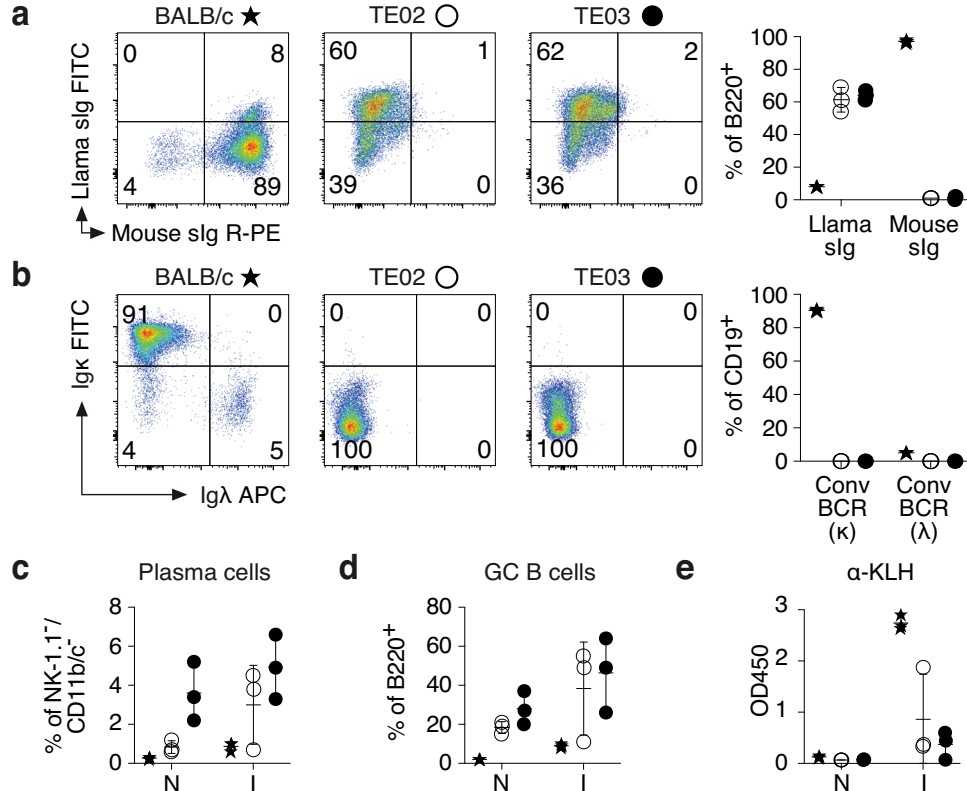

**Fig. 2 | LamaMice support the development of B cells that display llama heavy chain immunoglobulins. a**, **b** Spleen cells of 16-27-week-old mice (*n* = 3 per group) were stained with fluorochrome-conjugated antibodies and analyzed by flow cytometry. Gating was performed on B220[+] (**a**) or CD19[+] (**b**) cells. **c–e** 12-13-week-old mice (*n* = 3 per group) were immunized (I) with keyhole limpet hemocyanin (KLH). Naive (N) mice served as controls. **c**, **d** Four days after the second boost, spleen cells were stained with fluorochrome-conjugated antibodies and analyzed by flow cytometry. **c** Gating was performed on NK-1.1/CD11b/CD11c triple-negative cells. Plasma cells were identified as CD138[+]/PC-1[+] cells. **d** Gating was performed on

B220[+] cells. Germinal center (GC) B cells were identified as CD95[+]/surface immunoglobulin (sIg)[int] cells. **e** Serum was analyzed for the presence of KLH-specific antibodies (α-KLH) by ELISA using peroxidase-conjugated mouse Ig-specific (BALB/c) or llama Ig-specific (LamaMice) secondary antibodies. **a–e** Asterisks indicate samples from BALB/c mice, open circles from TE02 LamaMice, closed circles from TE03 LamaMice. **a**, **b** Dot plots are from single representative animals. Numbers indicate the % of cells in the respective quadrant. **a–e** Bar diagrams show the corresponding results for all mice in a group. Data represent mean ± SD for *n* = 3 individuals.

representative of each of the most commonly used llama VHH groups[40] (Supplementary Fig. 1); ii) to abrogate pairing of heavy and light chains during B cell development, we deleted the CH1 exon of IgM; iii) to allow class switch to a heavy chain-only IgG isotype, we replaced the IgD pseudogene with the IgG2b locus; iv) to enhance immunoglobulin synthesis, class switching, and somatic hypermutation[41], we inserted core elements of the 3' mouse locus control region; and v) to promote transmembrane signaling, we replaced the exons encoding the membrane-proximal, transmembrane, and cytosolic domains of llama IgM with the corresponding elements of the mouse. These modifications yielded BACs TE02 (Fig. 1b) and TE03 (Fig. 1c).

### Transfer of BACs into IgH-deficient mice restores B cell development

The linearized BACs were injected into the pronucleus of fertilized mouse oocytes. Progeny that transmitted the complete transgene were identified by PCR (Supplementary Fig. 2a) and backcrossed to B cell-deficient mice[42]. Flow cytometry analyses of peripheral blood cells showed that the transferred BACs restored the development of mature circulating B cells (CD19[+]) (Supplementary Fig. 2b). We named these new strains of llama hcAb-producing mice LamaMice.

### Cell surface and secreted antibodies of LamaMice contain only heavy chains

The results of further flow cytometry analyses showed that bone marrow and spleen of LamaMice contained all major subpopulations

of B220[+]/CD19[+] B cells, albeit at slightly different proportions than wildtype mice (Fig. 2 and Supplementary Figs. 3, 4), i.e. lower fractions of pro-B cells (CD43[+]/sIg[-]), immature B cells (CD43[-]/sIg[+]) and follicular B cells (FO, CD23[+]/CD21[low]), but higher fractions of immature B cells (IM, CD23[-]/CD21[-]) and marginal zone B cells (MZ, CD23[low]/CD21[+]). Importantly, splenic B cells of LamaMice display llama immunoglobulins on the cell surface, but not mouse-heavy or light chains (Fig. 2a, b). Interestingly, LamaMice contain a higher proportion of plasma cells (CD138[hi]/PC-1[hi]) and of germinal center B cells (GC B cells, CD95[+]/sIg[low]) than wildtype mice (Fig. 2c, d and Supplementary Fig. 4).

Immunization of LamaMice with keyhole limpet hemocyanin (KLH), a classic large protein antigen, resulted in expansion of these cell populations in both wildtype and LamaMice (Fig. 2c, d and Supplementary Fig. 4), as well as in KLH-specific serum antibody responses (Fig. 2e). Western-blot analyses of serum antibodies confirmed the presence of circulating hcAbs and the absence of conventional antibodies in LamaMice (Supplementary Fig. 2c).

### Nanobody discovery from LamaMice by hybridoma technology

An advantage of antibody discovery from mice over camelids is the availability of classical hybridoma technology. To assess the suitability of hybridoma fusions for nanobody discovery from LamaMice, we immunized LamaMice with non-replicative AAV8 (Fig. 3). AAV8 has a simple capsid composed of 60 monomers and is broadly used in gene therapy. Three days after the last boost, total cells from spleen and lymph nodes were fused to mouse Sp2/0 myeloma cells using a

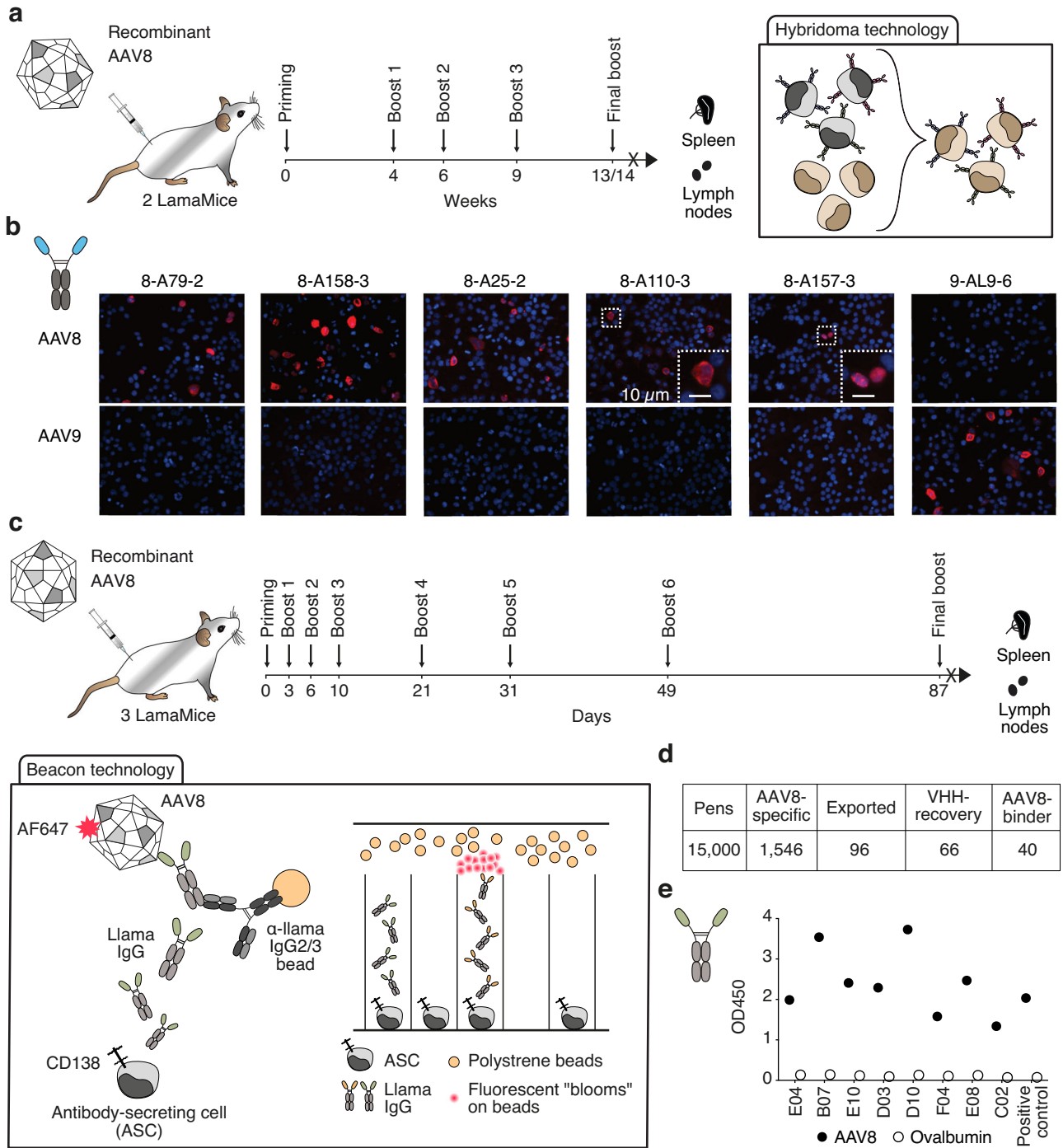

**Fig. 3 | Discovery of AAV-specific nanobodies from immunized LamaMice using classic hybridoma or single B cell screening technologies. a** LamaMice were immunized with AAV8 as indicated. Three days after the final boost, spleen and lymph node cells were fused with Sp2/0 mouse myeloma cells and cultured on 96-well plates in HAT-selection medium. Supernatants of hybridoma clones were screened by ELISA for reactivity with AAV8. Positive hybridomas were subcloned by limiting dilution. **b** The VHH-encoding region of positive clones was PCR-amplified, sequenced, and cloned into a mammalian expression vector upstream of the hinge, CH2 and CH3 of rabbit IgG. VHH-rabbit IgG hcAbs were produced in transiently transfected HEK-6E cells and cell supernatants were analyzed for reactivity with HEKAAV cells producing AAV by immunofluorescence microscopy. Bound hcAbs were detected with PE-conjugated anti-rabbit IgG. **c**, **d** LamaMice were immunized with AAV8 as indicated. Four days after the final boost, antibody-secreting cells (ASC) were sorted from spleen and lymph node cells using anti-CD138-coated beads, and cells were loaded into individual pens on a Berkley Lights Beacon® chip. Llama IgG heavy chain antibodies (hcAbs) captured on beads were detected with AF647-conjugated AAV8. Cells yielding an AAV8-specific signal were exported onto a 96-well plate and subjected to cDNA synthesis. **e** PCR-amplicons obtained with a VHH-specific primer pair were sequenced and cloned into a mammalian expression vector. VHHs were re-expressed as recombinant llama hcAbs and tested for specific binding to AAV8 by ELISA. Bound hcAbs were detected with PO-conjugated anti-llama IgG.

**Table 1 | Target-specific nanobody families derived from immunized LamaMice**

| F# | clone | Strain | Target | Var | SHM | V | D | J | CDR3 | K$_D$ (nM) |
|---|---|---|---|---|---|---|---|---|---|---|
| 1 | 8-A112-1 | TE02 | AAV8 | 3 | 3–8 | 5 | 2 | 6 | 14 | nd |
| 2 | 8-A158-3 | TE02 | AAV8 | 5 | 5–8 | 5 | 6 | 4 | 13 | nd |
| 3 | 8-A25-2 | TE02 | AAV8 | 1 | 10 | 2 | 6 | 4 | 13 | nd |
| 4 | 8-A110-3 | TE02 | AAV8 | 1 | 6 | VH | 7 | 4 | 7 | nd |
| 5 | 8-A157-3 | TE02 | AAV8 | 3 | 3–4 | 5 | 6 | 4 | 14 | nd |
| 6 | E04 | TE03 | AAV8 | 5 | 4–11 | 5 | 6 | 4 | 14 | nd |
| 7 | B07 | TE03 | AAV8 | 5 | 5–9 | 5 | 6 | 4 | 14 | nd |
| 8 | E10 | TE03 | AAV8 | 1 | 8 | 5 | 2 | 6 | 14 | nd |
| 9 | D03 | TE03 | AAV8 | 1 | 15 | 5 | 1 | 4 | 13 | nd |
| 10 | D10 | TE03 | AAV8 | 3 | 0–3 | 5 | 2 | 6 | 13 | nd |
| 11 | F04 | TE03 | AAV8 | 5 | 6–11 | 5 | 6 | 4 | 13 | nd |
| 12 | E08 | TE03 | AAV8 | 1 | 11 | 6 | 6 | 4 | 12 | nd |
| 13 | C02 | TE03 | AAV8 | 4 | 8–11 | 5 | 6 | 4 | 10 | nd |
| 1 | 8SG-G1 | TE02 | Spike-αβδ- | 3 | 1–3 | 3 | 6* | 4 | 14 | nd |
| 2 | 6SG-G12 | TE02 | Spike-αβδo | 14 | 2–7 | 5 | 7 | 6 | 3 | nd |
| 3 | 7SG-G9 | TE02 | Spike-α--o | 1 | 4 | 5 | 6 | 6 | 3 | nd |
| 1 | VHH398 | TE02 | hIgE | 4 | 4–6 | 5 | 1 | 7 | 13 | nd |
| 2 | VHH428 | TE03 | hIgE | 2 | 3 | 6 | 6 | 7 | 15 | 0.7 |
| 3 | VHH363 | TE02 | hIgE | 2 | 1 | 5 | 6 | 6 | 11 | 86 |
| 4 | VHH366 | TE03 | hIgE | 1 | 9 | 5 | 6 | 4 | 12 | 2 |
| 5 | VHH367 | TE03 | hIgE | 1 | 6 | 6 | 6 | 7 | 15 | 52 |
| 6 | VHH374 | TE02 | mIgE | 7 | 1–12 | 5 | 8 | 7 | 11 | 9.7 |
| 7 | VHH375 | TE02 | mIgE | 42 | 3–11 | 5 | 5 | 4 | 6 | 7.8 |
| 1 | A10 | TE03 | mIgG2c | 4 | 6–11 | 5 | 2 | 4 | 7 | 66 |
| 2 | C10 | TE03 | mIgG2c | 1 | 9 | 5 | 2 | 7 | 12 | 4 |
| 1 | A06 | TE02 | hCLEC9a | 1 | 2 | 5 | 1 | 4 | 7 | nd |
| 2 | A08 | TE03 | hCLEC9a | 3 | 2–3 | 5 | 3 | 4 | 7 | nd |
| 3 | A17 | TE03 | hCLEC9a | 8 | 1–3 | 5 | 6 | 4 | 8 | nd |

Nanobodies that share the same or a highly similar complementarity-determining region 3 (CDR3) were grouped into families and assigned an arbitrary family number (F#) and clone name. Var indicates the number of distinct nanobody sequences within a family. SHM indicates the range in the number of somatic hypermutations that distinguish members of a VHH family from the germline sequence of the parental V element. V indicates the parental V element used (see Supplementary Fig. 1); D and J indicate the respective elements identified by IMGT[60] or by visual inspection (marked with an asterisk). CDR3 indicates the number of amino acid residues of the CDR3. K$_D$ indicates the affinity determined by BLI (nd not determined).

standard polyethylene glycol fusion protocol (Fig. 3a). Upon cultivation in 96-well plates in HAT-selection medium, supernatants of growing clones were screened for the presence of AAV-specific llama antibodies by enzyme-linked immunosorbent assay (ELISA) using llama Ig-specific secondary antibodies. Positive clones were subcloned by limiting dilution.

Hybridoma cells from LamaMice offer easy access to antibody sequences using a single set of PCR primers. The deduced sequences allowed the grouping of the positive clones into distinct VHH-families, whose members share the same or highly similar complementary-determining region (CDR) 3 and framework regions (FRs) (Table 1).

We next cloned the VHH-encoding sequence of positive hybridomas into a mammalian expression vector upstream of the coding sequences for the hinge, CH2, and CH3 domains of rabbit IgG (Supplementary Fig. 5). Transient transfection of these plasmids into HEK-6E cells[43] grown in serum-free medium provided high yields of secretory hcAbs within six days after transfection (Supplementary Fig. 5a). We monitored the specificity of selected VHH-rabbit IgG hcAbs for

specific reactivity with AAV8 by immunofluorescence microscopy of HEK293AAV cells 48 hours after transfection with plasmids encoding AAV2, AAV8, or AAV9 (Fig. 3b and Supplementary Fig. 5b-d). We found that four VHH-families specifically recognized AAV8, and one VHH-family recognized both AAV8 and AAV2.

## Nanobody discovery from LamaMice by screening of single cells

The Beacon platform has enabled rapid antibody discovery from primary antibody-secreting cells (ASCs)[44]. ASCs are microfluidically sequestered in individual nanopens and screened for the production of specific antibodies using a bead-based binding assay that produces a characteristic fluorescent bloom. Individual positive cells are then exported into 96-well plates for amplification, cloning and sequencing of the VH and VL domains in case of conventional antibodies and of the VHH domains in case of heavy chain antibodies. To assess the suitability of the Beacon platform for nanobody discovery from LamaMice, we enriched ASCs from AAV8-immunized LamaMice based on CD138 surface expression (Fig. 3c). Fifteen thousand ASCs were loaded into nanopens of Beacon chips. Secreted llama hcAbs were captured on in-house developed assay beads. AlexaFluor647-conjugated AAV8 was used to detect pens containing ASCs secreting specific hcAbs. Approximately 10% (1546) of ASCs yielded AAV8-specific signals (Fig. 3d). Of these, 96 were exported into a 96-well plate containing lysis buffer. PCR-amplification with a VHH-specific primer pair yielded a product of the correct size from 66 wells, corresponding to ~70% PCR recovery. PCR recovered amplicons were cloned into a mammalian expression vector, re-expressed as recombinant llama heavy chain antibodies and tested for specific binding to AAV8 by ELISA (Fig. 3e). Forty clones representing eight distinct VHH families showed specific reactivity with AAV8 (Table 1). The results underscore the suitability of the Beacon platform for nanobody discovery from LamaMice.

## Nanobody discovery from LamaMice by direct cloning technology

Having shown how easily the VHH-encoding region can be PCR-amplified from hybridoma cells and from single ASCs, we hypothesized that a similar strategy could be adapted for nanobody discovery from immunized LamaMice by directly cloning the VHH-repertoire from lymph node or spleen cDNA into a mammalian expression vector. To test this hypothesis, we immunized five LamaMice with the receptor-binding domain (RBD) of the SARS-CoV-2 spike protein of the Wuhan strain (Fig. 4 and Supplementary Fig. 6). Three days after the final boost, we PCR-amplified and cloned the VHH-repertoire from splenocyte cDNA into the pCSE2.5 vector[43] upstream of the hinge, CH2 and CH3 of rabbit IgG (Fig. 4a). Plasmid DNAs prepared from single *E. coli* colonies were sequenced and used to individually transfect HEK-6E cells in a 96-well format. Aliquots of the serum-free supernatants were harvested five days after transfection and analyzed by SDS-PAGE (Supplementary Fig. 6a). For most clones, a Coomassie-stained band at ~45 kDa, i.e. the expected size of VHH-rabbit IgG heavy chains, was readily detected.

We next tested the HEK-6E supernatants containing VHH-rabbit IgG hcAbs for binding to HEK293T cells transiently transfected with the SARS-CoV-2 spike protein by flow cytometry. Twenty-one reactive nanobodies were assigned to three distinct VHH families based on their shared CDR3 and FRs (Table 1). We further analyzed whether the selected VHH-rabbit IgG hcAbs could also recognize the spike protein of different SARS-CoV-2 variants (Alpha, Beta, Delta, Omicron BA.1) using a multiplex flow cytometry assay (Fig. 4b and Supplementary Fig. 6b). We found that VHH 6SG-G12 and ten other clones recognized all four variants of the spike protein, whereas VHH 8SG-G1 and two other clones recognized the spike protein of SARS-CoV-2 Alpha, Beta and Delta strains, but not that of the Omicron strain (Fig. 4b). The reactivity of VHH 8SG-G1 resembles that of two nanobodies (Nb12 and Nb30) previously selected from nanomice (Fig. 4b)[39] and that of a

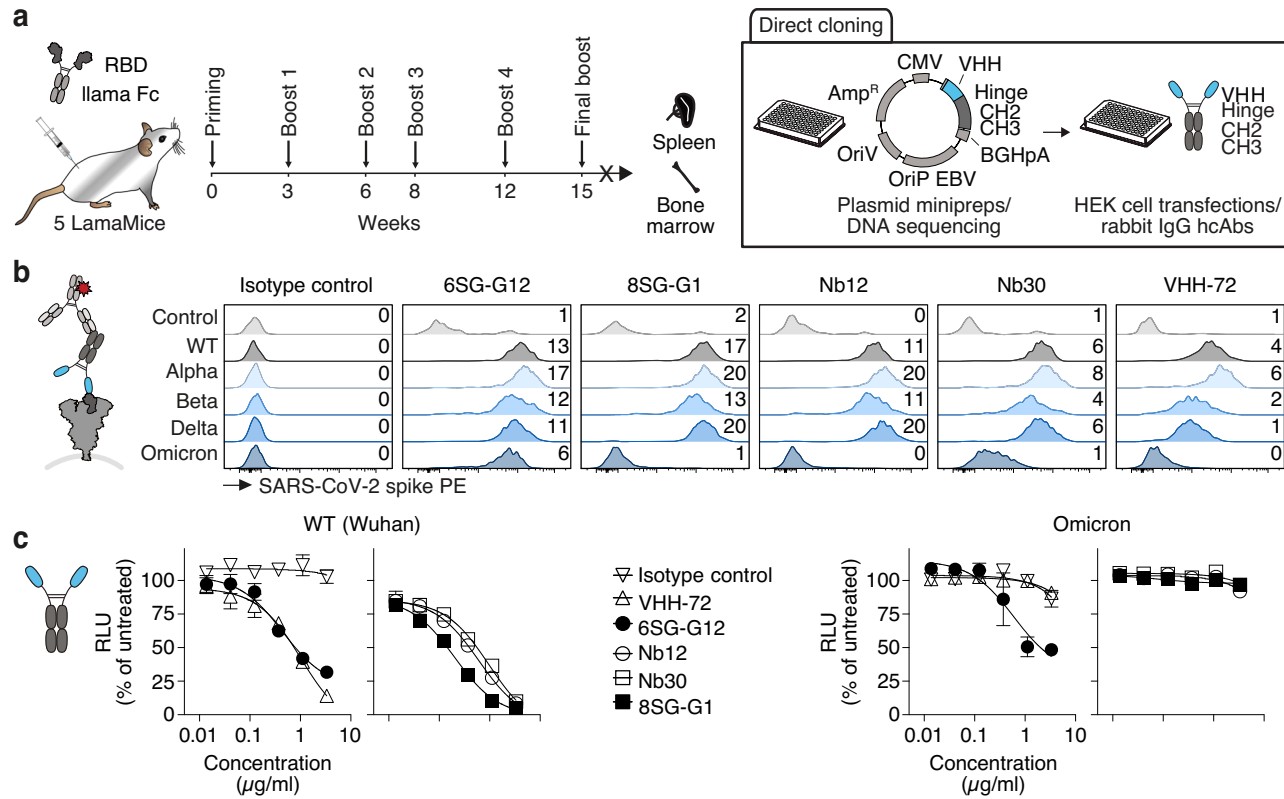

**Fig. 4 | Discovery of RBD-specific nanobodies from immunized LamaMice using direct cloning technology. a** LamaMice were immunized with a fusion protein comprising the recombinant receptor-binding domain (RBD) of the SARS-CoV-2 spike protein fused to the hinge, CH2 and CH3 domains of llama IgG2b. Three days after the final boost, the VHH-repertoire was PCR-amplified from cDNA prepared from spleen and bone marrow cells. PCR-amplicons were cloned into the pCSE2.5 expression vector upstream of the hinge, CH2 and CH3 domains of rabbit IgG. Plasmid DNA was prepared from individual colonies grown in 96-well plates, sequenced, and transiently transfected into HEK-6E cells cultivated in serum-free medium in 96-well plates. Supernatants were harvested 5 days after transfection. A liquots of the supernatants were analyzed for the production of heavy chain antibodies (hcAbs) by SDS-PAGE (see Supplementary Fig. 6a). **b** VHH-rabbit IgG hcAbs in HEK-6E supernatants were screened for binding to HEK293T cells transiently co-transfected with expression vectors for GFP and the spike protein of various SARS-CoV-2-strains. Bound antibodies were detected with PE-conjugated anti-rabbit IgG. Numbers indicate the mean PE fluorescence intensities (MFI x 10⁻³) of the GFP⁺ cell population. Parallel stainings were performed with VHH-rabbit IgG hcAbs containing nanobodies discovered by other groups from nanomice immunized with recombinant RBD (Nb12, Nb30) and a llama immunized with the spike protein of SARS-CoV-1 (VHH-72). **c** HEK293T cells stably overexpressing human ACE2 were incubated with luciferase-encoding lentiviral gene ontology vectors pseudotyped with SARS-CoV-2 spike protein of the wild type (WT) (Wuhan) or Omicron BA.2 variant in the presence of titrated amounts of the indicated VHH-rabbit IgG hcAbs. Two days after transduction, luciferase activity was quantified on a luminometer, 20 min after addition of luciferin. Data represent mean ± SD for triplicates.

nanobody (VHH-72) from a llama immunized with the spike protein of SARS-CoV-1[45]. We further assessed the capacity of representative VHH-rabbit IgG hcAbs to prevent infection of HEK293T cells stably over-expressing human ACE2 by luciferase-encoding lentiviral vectors pseudotyped with the spike protein of the Wuhan wild type or Omicron BA.2 variant. The results show that all analyzed VHH-rabbit IgG hcAbs were able to neutralize the Wuhan strain. VHH 6SG-G12 also neutralized the Omicron BA.2 variant (Fig. 4c). Together, these experiments demonstrated the potential of LamaMice to generate functional antibodies neutralizing SARS-CoV-2 pseudoviruses in vitro, underscoring their potential use as therapeutic antibodies.

**Nanobody discovery from LamaMice by phage display technology**

Nanobody discovery from immunized camelids usually occurs via phage display of the VHH-repertoire amplified from peripheral blood leukocytes (PBLs) of immunized animals. To evaluate the suitability of phage display technology for nanobody discovery from LamaMice and to perform a side-by-side comparison of nanobodies obtained from immunized LamaMice and alpacas, we immunized two alpacas and six LamaMice with the same cocktail of mouse and human IgE as target antigen (Fig. 5). Three days after the last boost, the VHH-repertoire was

PCR-amplified from PBL cDNA of the alpacas and from spleen and lymph node cDNA of the immunized LamaMice and cloned into the pHEN2 phage display vector (Fig. 5a). Selection of target-specific nanobody-displaying phages was carried out by two rounds of solution panning on biotinylated IgE and capture on streptavidin beads. Enriched clones were sequenced and grouped into families based on similar CDR3 and framework sequences. Seven distinct families were identified from immunized LamaMice and five families from the immunized alpacas.

Candidate VHHs were subcloned into the pCSE2.5 expression vector upstream of the hinge, CH2 and CH3 of rabbit IgG, and produced as secretory proteins in transiently transfected HEK-6E cells. VHH-rabbit IgG hcAbs were evaluated for specific binding to mouse and human cell lines expressing cell surface IgE by flow cytometry (Fig. 5b). The results show that VHH428 and four other VHHs bind specifically to human IgE-expressing U-266 myeloma cells, but not to mouse IgE-expressing A174 or mouse IgG1-expressing B64 hybridoma cells. Conversely, VHH374 and VHH375 showed exclusive binding to A174 cells (Fig. 5b).

Biochemical properties of recombinant monomeric VHHs from LamaMice and alpacas were further analyzed by nano-differential scanning fluorimetry to determine their thermal stability and

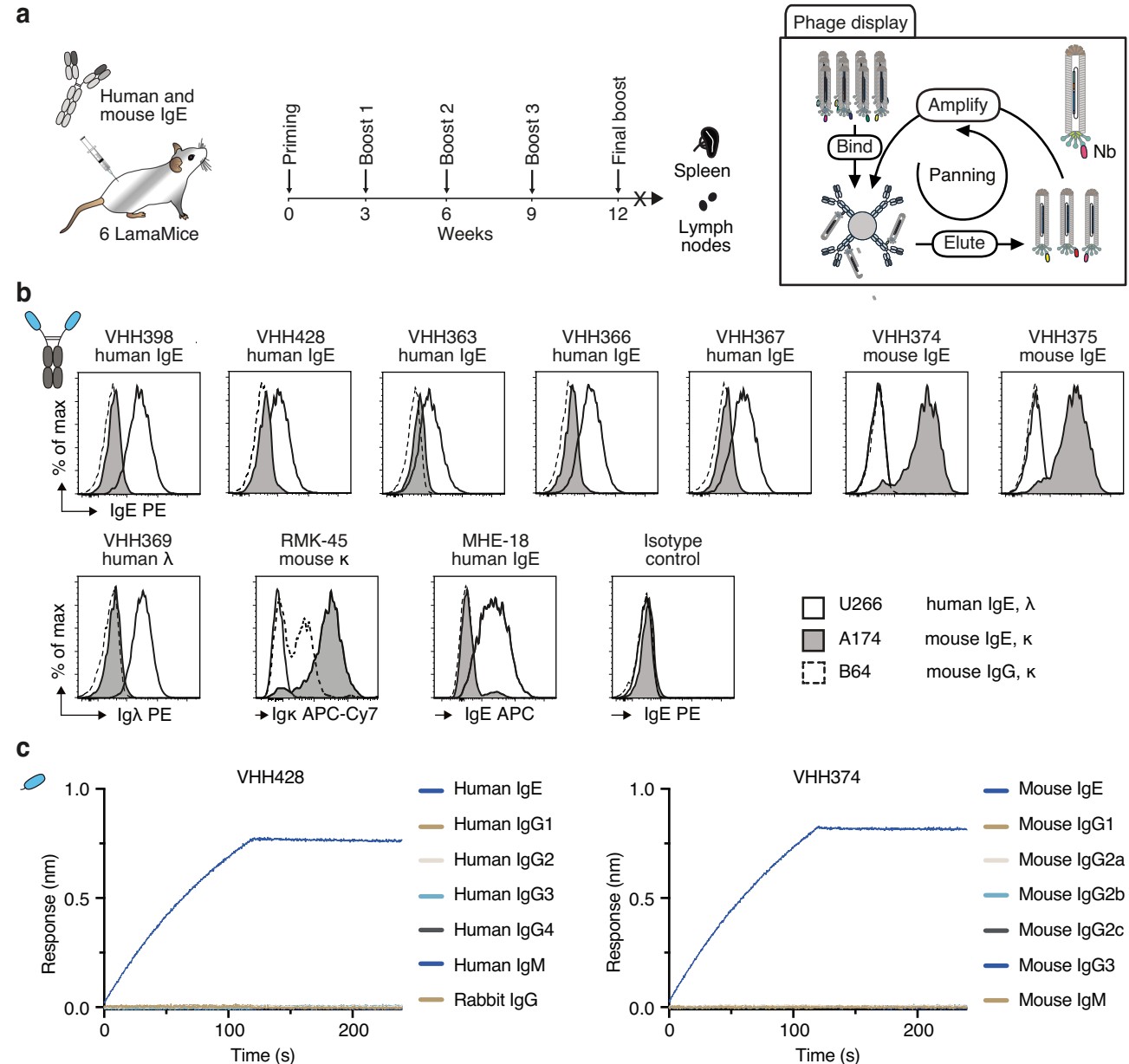

**Fig. 5 | Discovery of IgE-specific nanobodies from immunized LamaMice using phage display technology. a** LamaMice were immunized with a cocktail of human IgE and mouse IgE. Three days after the final boost, the VHH-repertoire was PCR-amplified from cDNA prepared from spleen and lymph nodes. PCR-amplicons were cloned into the pHEN2 phagemid vector upstream of the coding sequences for a His6x-c-Myc tag, an amber stop codon followed by the region encoding the gp3 surface protein of the M13 phage. Phage display libraries prepared from transformed *E. coli* were panned in solution on biotinylated IgE and complexes were captured on streptavidin beads. **b** The VHH-encoding region of enriched phages was sequenced and subcloned into the pCSE2.5 expression vector upstream of the hinge, CH2 and CH3 domains of rabbit IgG as in Fig. 4a. Specific binding of VHH-rabbit IgG hcAbs to cell lines expressing B cell receptors composed of human IgE (myeloma U-266), mouse IgE (hybridoma M1-3A174), or mouse IgG2a (M261B64) was analyzed by flow cytometry. Bound antibodies were detected with PE-conjugated anti-rabbit IgG. Control stainings were performed with anti-human Igλ (VHH369-rabbit IgG hcAb) or a VHH-rabbit IgG hcAb isotype control followed by PE-conjugated anti-rabbit IgG, and with directly conjugated anti-mouse Igκ (RMK-45) or anti-human IgE (MHE-18). **c** Nanobodies VHH428 and VHH374 were biotinylated and captured on streptavidin-coated biosensors. Specific binding to IgE and the other indicated Ig isotypes was analyzed by biolayer interferometry.

aggregation behaviour and by biolayer interferometry (BLI) to estimate their affinity (Supplementary Fig. 7). The results show that nanobodies from LamaMice and nanobodies from alpacas display a similar range of biochemical properties.

The specificity of VHH428 and VHH374 towards different immunoglobulin isotypes was further analyzed by BLI using biotinylated immunoglobulin isotypes immobilized on streptavidin-coated biosensors (Fig. 5c). The results confirm the high specificity of these nanobodies for human and mouse IgE, and the lack of binding to any other immunoglobulin isotypes (Fig. 5c). These two nanobodies are now commercially available from ChromoTek as Nano-CaptureLigand human IgE (VHH428) and Nano-CaptureLigand mouse IgE (VHH374). In a separate nanobody discovery campaign, we selected mouse IgG2c-specific nanobodies from immunized LamaMice using the direct cloning strategy (Supplementary Fig. 8).

### Nanobody discovery from cDNA-immunized LamaMice

We previously demonstrated the capacity to select membrane protein-specific nanobodies from llamas and alpacas immunized with a cDNA expression vector encoding the target antigen[31,46]. To

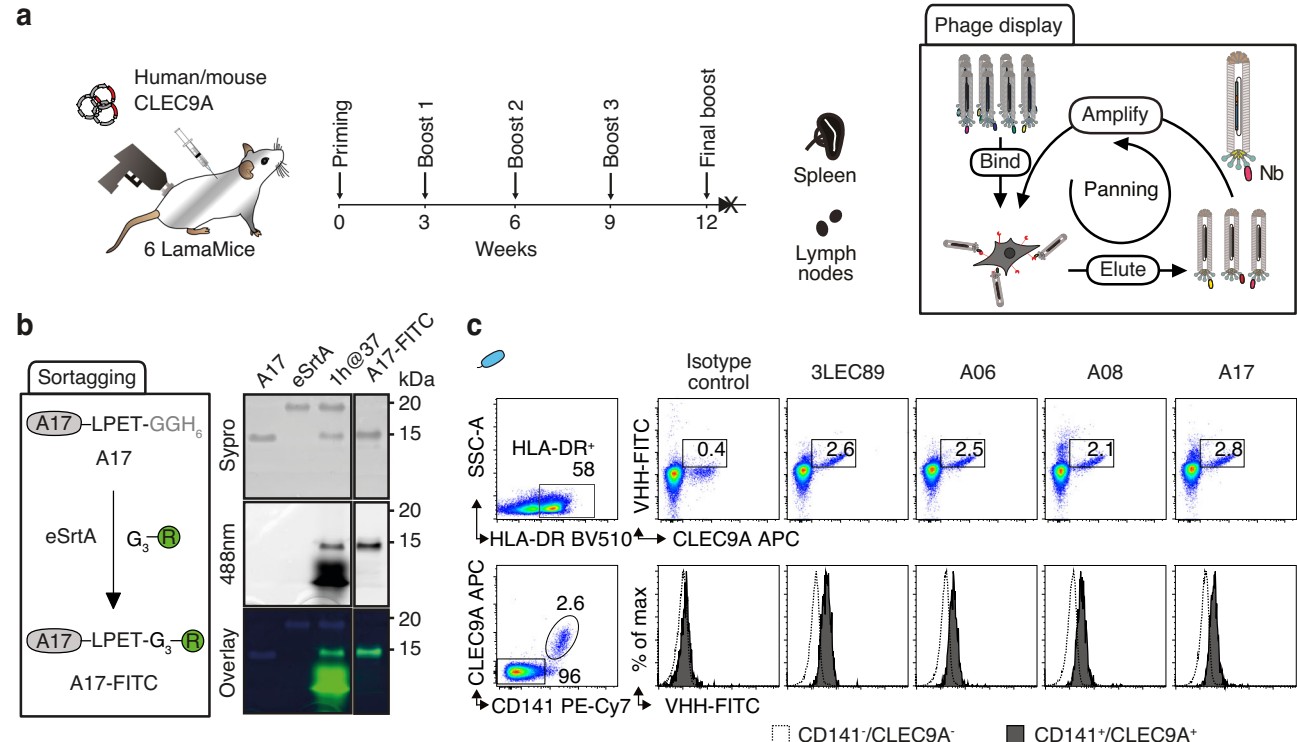

**Fig. 6 | Discovery of CLEC9A-specific nanobodies from cDNA-immunized LamaMice using phage display technology. a** LamaMice were immunized with a cDNA expression vector encoding human CLEC9A using a gene gun. Three days after the final boost, the VHH-repertoire was PCR-amplified from cDNA prepared from spleen and lymph nodes and cloned into the pHEN2 phagemid vector as in Fig. 5. Phage display libraries were panned on cells expressing cell surface CLEC9A. **b** The VHH-encoding region of enriched phages was subcloned upstream of the coding sequence for a sortase tag and VHHs were fused to a FITC-conjugated

peptide using sortase (eSrtA). Unreacted product and sortase were removed with Ni-NTA resin and excess nucleophile was removed by spin filtration. In-gel Sypro (protein) and FITC fluorescence of reaction products were analyzed via SDS-PAGE. **c** Binding of FITC-conjugated VHHs to CLEC9A-expressing human peripheral blood cells was analyzed by flow cytometry. Gating was performed on MHCII HLA-DR+ cells (upper panels). Cells were counterstained with commercial CLEC9A-specific and CD141-specific mAbs. Gating was performed on HLA-DR+ cells that were either CD141+ or CD141- (lower panels).

evaluate the suitability of cDNA immunization technology for nano-body discovery from LamaMice, we immunized six LamaMice with a cDNA expression vector encoding the human C-type lectin receptor CLEC9A (Fig. 6). Even though serum prepared from these mice did not show any detectable CLEC9A-specific antibodies, we PCR-amplified the VHH-repertoire from spleen and lymph node cDNA and cloned it into the pHEN2 phage display vector (Fig. 6a). Selection of target-specific nanobody-displaying phages was carried out by panning on human CLEC9A-transfected HEK293T cells. Enriched clones were sequenced and grouped into VHH-families. Three distinct families were identified from immunized LamaMice (Table 1). The VHH-encoding region of a representative member of each family was subcloned into the pCSE2.5 vector and expressed as a secreted VHH-rabbit IgG hcAb. Flow cytometry analyses confirmed the specific binding of all three VHHs to human and cynomolgus CLEC9A, but not to mouse CLEC9A (Supplementary Fig. 9). To allow site-specific labeling of the recombinant monovalent VHHs, we introduced a C-terminal five amino acid sortase recognition sequence during cloning of the VHH-encoding sequences into the pET22b(+) expression vector. Using sortase labeling technology[47], we fused the selected and control VHHs to a FITC-conjugated peptide (Fig. 6b), then evaluated their specific binding to human pan-dendritic cell-enriched peripheral blood mononuclear cells (Fig. 6c). CLEC9A has a restricted expression on cross-presenting CD141+ type-1 conventional dendritic cells (cDC1)[48]. The results show that the three selected VHHs bind specifically to CLEC9A-expressing cells, with the same efficiency as VHH 3LEC89(+), a nanobody previously selected from a CLEC9A-immunized alpaca[49].

## Discussion

The high potential of nanobodies is illustrated by their recent use in new innovative biotechnological tools and therapeutics that harness the high stability of the nanobody scaffold. The new mouse model described in this study greatly facilitates the discovery of target-specific nanobodies. In the design of LamaMice, we aimed to eliminate known difficulties with hcAb responses in Ig-tg mice based on human VH elements. We achieved this by using llama VHH genes, resulting in an animal model that does not produce any competing tetrameric antibodies. Upon immunization, LamaMice generated specific llama hcAbs against a variety of antigens. We demonstrate that antigen-specific nanobodies can be recovered from immunized LamaMice by established hybridoma, single B cell screening, and phage display technologies. Moreover, we show that affinity-matured nanobodies can readily be recovered by directly cloning the VHH-repertoire into a mammalian expression vector. In each case, the sequence of the nanobody was readily available, and high yields of soluble, stable nanobodies and hcAbs were obtained at high purity from the supernatants of transiently transfected HEK-6E cells cultured in serum-free medium. Side-by-side analyses of nanobodies obtained from Lama-Mice and alpacas underscore the high quality of nanobodies obtained from LamaMice. A key advantage of nanobodies over the paired VH and VL domains of conventional antibodies is that they can be readily converted into monovalent, bivalent, and multivalent formats (Supplementary Fig. 10). Nanobodies derived from LamaMice can be converted just as easily as nanobodies from alpacas into any such format.

Compared to single-domain antibodies from synthetic libraries[33–35], nanobody discovery from immunized LamaMice profits

from clonal expansion and affinity maturation of the natural immune response[31]. Compared to camelids, LamaMice produce only VHH-based hcAbs, thus avoiding a competing conventional antibody response[23]. LamaMice further offer the advantage of genetic technologies. For example, LamaMice can be crossed with target-deficient mouse lines or target genes could be deleted in LamaMice by CRISPR/Cas or similar technologies. Provided that the physiological antibody response is not compromised by the defective target gene, immunization of such target-deficient LamaMice with the human gene product is expected to yield antibodies also against conserved epitopes[50]. Compared to human VH3-based transgenic mice[36–38], LamaMice have the advantage of a repertoire consisting of camelid VHH framework sequences that have been shaped by millions of years of evolution for high stability and solubility. This reduces the risks of association with light chains and the risk of aggregation, thereby allowing easier development of clinical candidates.

Potential limitations of LamaMice compared to llamas and other camelids are a comparatively smaller B cell-repertoire of the smaller mammal and a weak antibody response to cDNA-immunization. The former can be compensated by immunizing a larger number of mice. The latter might be improved by co-administration of an expression vector encoding a proinflammatory cytokine or a checkpoint inhibitor and/or by genetic fusion of the target to potent T cell epitopes[31,51]. Considering that homozygosity at the BAC integration site may impair health, it is prudent to use heterozygous LamaMice for nanobody discovery.

LamaMice are based on a modular transgene platform that allows further modifications by BAC recombineering, i.e. without the need to synthesize a completely new transgene each time[52]. LamaMice thus combine powerful technologies that facilitate nanobody discovery for broad applications in biotechnology and medicine.

## Methods

### Animal experiments
Animal experiments were performed according to national and institutional animal care and ethical guidelines and were approved by the Veterinarian Agency of Hamburg and the local animal care committee (registration number A029/2019). All mice were maintained in a specific-pathogen-free facility at temperatures of 21–24 °C with 40-70% humidity on a 12 h light/12 h dark cycle and provided with food and water *ad libitum*. When applicable, animal experiments were conducted in accordance with the ARRIVE 2.0 guidelines. The number of animals used for experiments is specified in the respective figure legends. As these studies were largely exploratory in character, no sample size was calculated a priori. No specific inclusion or exclusion criteria applied. No animals and no data points were excluded from the analyses. Animals were randomized if feasible (e.g. for the comparison of non-immunized with immunized mice from the same genetic background). Investigators were blinded for biological analyses (e.g. flow cytometry) and data analyses.

### Isolation of clones covering the llama IgH locus from a BAC-library
The liver of a llama, housed and sacrificed in accordance with the institutional animal welfare guidelines, was kindly provided by Matthias Gauly, Dept. of Animal Sciences, University of Göttingen, Germany. Genomic DNA from this liver was partially digested with *HindIII* and cloned into the BAC vector pCC1 (Epicentre), yielding a library with an average insert size of ~150 kb. Radiolabeled probes encoding the VHH and the CH2-CH3 domains of IgG2b were used to identify 17 hybridizing BACs with insert sizes of 90–220 kb. Southern Blot analyses identified two overlapping BACs carrying variable (V03, ~120 kb) and constant segments (F07, ~100 kb) of the immunoglobulin heavy chain (IgH) gene locus. The inserts were sheared into 5 kb fragments and sequenced by Sanger technology. Gaps were filled by primer-

walking and full sequences were assembled using Lasergene software. The sequences were deposited in GenBank (JQ684648, JQ684647).

### BAC recombineering and generation of llama IgH transgenes
BAC V03 was genetically modified using BAC recombineering technology. DNA fragments were generated either by PCR using primers carrying 50 bp overhangs homologous to the respective insertion sites or as synthetic DNA fragments containing flanking homologous overhangs of 80–270 bp (Supplementary Table 1). All modifications were verified by analytical restriction digestion and DNA sequencing. Regions HS3b and HS4 of the mouse LCR (alpha-enhancer, αE) were PCR-amplified from a plasmid (kindly provided by Michel Cogné, Limoges, France)[41] and inserted into the 3′-end of BAC V03 using the Counter Selection BAC Modification Kit (GeneBridges). All subsequent steps were performed with the recombineering strain *E. coli SW106* (kindly provided by Neal Copeland, Frederick, MD, USA)[53]. The Cδ pseudogene was replaced by a spectinomycin resistance cassette. The endogenous loxP site of BAC V03 was replaced by a tetracycline resistance cassette. The CH1 domain of the Cμ locus was replaced by a neomycin resistance cassette flanked by the Lox inverted repeat variants lox71 and lox66[52]. After homologous recombination, Cre/loxP recombination was used to delete the neomycin selection cassette, yielding a remnant lox72 mutant. The spectinomycin selection cassette was replaced by a 13.9 kb fragment covering the *Lama glama* Cγ2b locus. For this, a Cγ2b fragment was cloned from BAC F07 into pBluescript II KS (+)/LIC. A synthetic DNA fragment containing a 5′-homology arm of 285 bp size and a *NruI* restriction site was cloned into the *ClaI* site upstream of the Cγ2b locus. Another synthetic DNA fragment encompassing an ampicillin resistance cassette flanked by the loxP mutants lox71 and lox66 fused to a 270 bp 3′-homology arm, and a second *NruI* restriction site was cloned into the *BstZ17I* site downstream of the Cγ2b locus. The insert was recovered by digestion with *NruI*. After homologous recombination, Cre/loxP recombination was used to delete the ampicillin selection cassette, yielding a remnant lox72 mutant. A synthetic DNA fragment containing an ampicillin selection cassette followed by elements HS3a and HS1,2 of the mouse 3′ locus control region was flanked by the two loxP mutants lox71 and lox66, 86–90 bp homology arms and a *SnaBI* restriction cleavage site. The DNA cassette was isolated by *SnaBI* restriction digestion. After homologous recombination, Cre/loxP recombination was used to delete the ampicillin selection cassette, yielding a remnant lox72 mutant. A synthetic DNA fragment encoding five additional V$_H$H exons within a scaffold of mouse IgH-V locus sequences from functional, highly transcribed, and often used V-elements, i.e. IgH-V promoter regions 186.2 and 17.2.25 or combinations thereof, an ampicillin selection cassette flanked by lox71 and lox66 sites, and 390 bp homology arms containing *NruI* restriction sites was cloned into pUC57. The exon for the leader peptide, the downstream intron, and the recombination signal sequence (RSS) at the 3′-end of the V$_H$H exon were retained in the camelid configuration. Repetitive DNA and remnant viral elements were omitted to minimize the size of the construct. The V$_H$H cassette was isolated via *NruI* restriction sites encoded in the homology arms. After homologous recombination, Cre/loxP recombination was again used to delete the ampicillin selection cassette. This modified BAC was designated TE02.

This BAC was further modified to replace the region encoding the membrane-proximal and transmembrane domains of llama IgM by corresponding sequences of mouse IgM. For this, a synthetic DNA fragment encompassing the genomic sequence of the mouse IgM locus from the CH3 exon to the TM2 exon was flanked with 45 bp homology regions, an ampicillin selection cassette flanked by lox71 and lox66 inserted into the intron between exons CH4 and TM1, 45 bp homology arms and *SnaBI* restriction sites was cloned into the pUC57. The insert was recovered by *SnaBI* digestion. After homologous

recombination, Cre/lox recombination was used to delete the ampicillin selection cassette, yielding transgene TE03.

## Generation and maintenance of TE02 and TE03 LamaMice

Transgenes TE02 and TE03 were purified using the PhasePrep BAC DNA Kit (Sigma-Aldrich) followed by *Not*I digestion and phenol/chloroform extraction. The concentration of the linearized DNA was adjusted to 1 ng/µl. 1 µl was injected into the male pronucleus of fertilized eggs of C57BL/6 J x CBA (B6/CBA) mice according to standard procedures and subsequently implanted into pseudopregnant recipients[54]. Founder mice were backcrossed to Ig-deficient JHT-mice (B6.129P2-*Igh-J*$^{tm1Cgn}$, kindly provided by Klaus Rajewsky, Berlin)[42]. Transgene transmission was verified by PCR from genomic ear clip DNA using Platinum blue PCR super mix (Invitrogen) and pairs of primers specific for the 5′- and 3′-ends of the transgene (Supplementary Table 1). Deletion of the J-elements and the µ-enhancer was verified using the diagnostic primer pair provided in the technical support protocol of Stock No: 002438 (The Jackson Laboratory). Two founders were identified for TE02, and four founders for TE03. The founders were not interbred. The number of circulating B cells was monitored by flow cytometry of a blood sample taken at 6-8 weeks of age (Supplementary Fig. 2b). For each line, we chose a founder whose offspring consistently showed a high number of circulating B cells. In order to transfer the intact BALB/c MHC locus (H2$^d$) to LamaMice, we backcrossed TE02, TE03, and JHT mice to BALB/c mice. The BAC structure and integration site in these founders were analyzed by Cergentis, Utrecht, NL. The results show that BAC TE02 integrated in the *Abca5* gene on Chromosome 11. A 24 kb genomic DNA fragment (chr11:110,311,269-110,335,224) encompassing exons 2-9 of *Abca5* was deleted at the integration site. BAC TE03 integrated downstream of the *Tmem232* gene on Chromosome 17. A 17 kb genomic DNA fragment (chr17:65,372,523- 65,645,975) encompassing an intergenic region was duplicated at the integration site. In both cases, DNA sequencing verified the presence of the entire transgene. TE02 and TE03 LamaMice are now maintained by interbreeding as hemizygous and homozygous for the BAC. Every 6-10 generations, the lines are backcrossed to JHT mice.

## Primary cells and cell lines

Single-cell suspensions were prepared from mouse bone marrow, spleen, lymph node, and peripheral blood. Erythrocytes were lysed with ACK lysis buffer in samples from blood, spleen, and bone marrow. Experiments with human blood were conducted according to the guidelines of the Dutch Medical Research Involving Human Subjects Act (WMO). Approval was granted by the Medical Ethical Committee Arnhem-Nijmegen (CMO). Written informed consent was obtained from all participants. Human dendritic cells were isolated from buffy coats of healthy blood donors (Sanquin). Mononuclear cells were purified by density gradient centrifugation on Lymphoprep (Stemcell), erythrocytes were lysed with ACK lysis buffer [150 mM NH$_4$Cl, 10 mM KHCO$_3$, 0.1 mM Na$_2$EDTA, pH 7.2–7.4]. Pan-dendritic cells (panDCs) were isolated by magnetic cell separation using the human panDC enrichment kit according to the manufacturer's instructions (Miltenyi Biotec; Bergisch Gladbach, German).

Mouse hybridoma cell lines producing mouse IgE (clone M1-3A174) and mouse IgG2a (clone 261-B64) were produced in house at the antibody core facility of the University Medical Center Hamburg Eppendorf. The human U-266 and LP-1 myeloma cell lines and HEK293T cells were obtained from the Leibniz-Institute DSMZ-German Collection of Microorganisms and Cell Cultures, Braunschweig, Germany. HEK293-6E cells were licensed from the National Research Council, Canada[55]. Expression constructs encoding mouse CLEC9A (Genbank NM_001205363.1), human CLEC9A (Genbank NM_207345.4), cynomolgus CLEC9A (Genbank NM_001194664.3), GFP fused to the nuclear translocation sequence of LKLF, the RBD or spike protein of

SARS-CoV-2 variants (Addgene #183413, 183417, 170449, 172320, 180843, 183700) (5 µg per 10$^6$ cells) were transfected into HEK293T cells using polyethylenimine (Q-Biogen)[46].

## Flow cytometry

Fc receptors were blocked with αCD16/32, and cells were stained with the indicated antibodies (Supplementary Table 2). Dead cells were excluded by staining with Pacific Orange or Alexa Fluor 750 succinimidyl esters (Invitrogen). For multiplex analyses, HEK293T cells transiently co-transfected with expression constructs for GFP and a particular Spike variant were prestained with eFluor 450 and/or eFluor 670 (Invitrogen) and washed. Prestained cells were then mixed and analyzed for binding to VHH-rabbit IgG hcAbs. Bound hcAbs were detected with PE-conjugated donkey anti-rabbit IgG (H + L) (Dianova). Cells were analyzed on a BD FACSCanto II, FACSCelesta, FACSLyric, or FACSymphony A1 flow cytometer using Diva (Becton Dickinson) and FlowJo (Treestar) software. Compensation of spectral overlap was performed with the same antibodies immobilized on UltraComp eBeads (Invitrogen).

## Immunizations

For comparative analysis of B cell activation, mice were immunized s.c. with 25 µg KLH in 200 µl of a 1:1 oil-in-saline emulsion with Sigma adjuvant system (SAS, 0.5 mg/ml Monophosphoryl Lipid A (detoxified endotoxin) from *Salmonella minnesota*, 0.5 mg/ml synthetic Trehalose Dicorynomycolate, 0.2% squalene-Tween 80) and i.p. with 25 µg KLH in 200 µl saline containing 50 µg polyinosinic:polycytidylic acid (poly I:C, Sigma). Two weeks after priming, mice received two boost immunizations with 15 µg KLH in SAS s.c. and 15 µg KLH in poly I:C i.p. in a two-week interval. Mice were sacrificed four days after the second boost. Plasma was prepared and analyzed for the presence of KLH-specific antibodies by ELISA and for mouse and llama immunoglobulins by Western blot analyses. Spleen and bone marrow cells were analyzed by flow cytometry.

For all Nb discovery campaigns, LamaMice were immunized as illustrated schematically in the corresponding figures. Subcutaneous (s.c.) injections of protein antigens with adjuvant were distributed at four sites in the upper and lower back and the hock, intravenous injections (i.v.) were administered in saline without adjuvant into the tail vein, and intraperitoneal injections (i.p.) with poly I:C were injected into the lower left quadrant of the abdomen. Mice were sacrificed 3-4 days after the final boost. Blood was obtained by heart puncture and used for serum or plasma preparation. Single-cell suspensions were prepared from the spleen, bone marrow, and lymph nodes. Cells were either directly fused with Sp2/0 myeloma cells or frozen after lysis of erythrocytes in aliquots of 5–10 × 10$^6$ cells in RA1 buffer (50% guanidinium thiocyanate, Macherey, and Nagel), 15 mM DTT for RNA and cDNA preparation.

For the AAV Nb discovery campaigns, AAV (1 × 10$^{11}$ viral genomes) were administered s.c. in 200 µl SAS emulsion and the same dose of AAV was administered i.p. in 150 µl PBS containing 50 µg poly I:C. Mice received four to six boost immunizations and were sacrificed three or four days after the last immunization.

For the RBD Nb discovery campaign, 15 µg RBD-llama IgG2b Fc fusion protein was administered s.c. in 200 µl SAS-emulsion and 15 µg antigen in 150 µl PBS containing 50 µg poly I:C was injected i.p. Mice received five boost immunizations in 2–4-week intervals. For the final boost immunization, 30 µg were distributed s.c. with SAS, i.p. with poly I:C and i.v. in saline.

For the IgE and IgG2c Nb discovery campaigns, 30 µg recombinant IgE or 30 µg MU1067-mIgG2c hcAb was administered s.c. in 200 µl of a 1:1 saline-in-oil emulsion with Complete Freund's Adjuvant (CFA, 85% paraffin oil, 15% mannide monooleate containing 0.1 mg/ml heat killed and dried *Mycobacterium tuberculosis* (H37Ra, ATCC 25177), Sigma). Mice received four boost immunizations in 3-week intervals with 30 µg

antigen s.c. in Incomplete Freund's Adjuvant (IFA, 85% paraffin oil, 15% mannide monooleate, Sigma). For the final boost, 30 μg IgE or 30 μg hcAb in 200 μl saline was administered s.c., i.p. and i.v.

For the CLEC9A Nb discovery campaign, mice were immunized intradermally by ballistic DNA immunization with 1 μm gold particles coated with cDNA expression vectors encoding mouse CLEC9A, human CLEC9A, and rat GM-CSF mixture using a Helios Gene Gun (Biorad) (for each immunization 5 shots with 1 μg DNA/mg gold)[31,46]. Mice received four ballistic DNA boost immunizations in 3-week intervals. For the final boost three days before sacrifice, mice were injected s.c., i.p., and i.v. with HEK-6E cells transiently transfected with expression vectors for mouse Clec9a and human CLEC9A.

### Western-Blot analysis of serum IgG
Immunoglobulins in plasma of LamaMice (10 μl per animal), BALB/c- (2.5 μl), and JHT-mouse (2.5 μl) were precipitated in duplicates with protein A immobilized on Sepharose beads. Bound proteins were eluted from washed beads with reducing SDS sample buffer (Invitrogen), size fractionated by SDS-PAGE, and transferred onto two PVDF membranes. One membrane was incubated with peroxidase-conjugated sheep anti-mouse IgG antibody (GE Healthcare, NA931), the other with peroxidase-conjugated goat anti-llama IgG (H + L) antibody (Bethyl Laboratories). Membranes were washed and incubated at RT with SuperSignal West Pico PLUS Chemiluminescent substrate (ThermoFisher Scientific). Images were acquired using the Molecular Imager ChemiDoc XRS System and Quantity One software (Bio-Rad Laboratories).

### ELISA
Humoral immune responses were monitored in diluted serum or plasma by ELISA on microtiter plates (Nunc MaxiSorp, Thermo Fisher Scientific, Waltham, MA) coated with 100 ng of KLH or AAV particles (containing $1 \times 10^9$ viral genomes). Bound antibodies were detected using peroxidase-conjugated goat anti-llama IgG (H + L) antibody (Bethyl Laboratories) or peroxidase-conjugated sheep anti-mouse IgG antibody (GE Healthcare, NA931), and 3,3',5,5'-tetramethylbenzidine (TMB) (Sigma) as substrate. The absorbance at 450 nm was measured using a Victor3 ELISA reader (Perkin-Elmer).

### cDNA amplification and cloning of the VHH-repertoire
Lymphocytes and hybridoma cells were lysed in RA1 buffer (Macherey-Nagel) with 15 mM DTT, and RNA was prepared using the NucleoSpin RNA kit (Macherey-Nagel). cDNA synthesis was performed with IgG2b- or IgM-specific primers (Supplementary Table 1). The VHH-encoding region was amplified by two consecutive PCRs, with primer pairs of the second PCR carrying appropriate restriction sites (Supplementary Table 1).

For the production of hcAbs in HEK cells, PCR products were cloned into the pCSE2.5 vector (kindly provided by Dr. Thomas Schirrmann, Braunschweig[43]) downstream of the IgH-leader peptide and upstream of the coding regions for the hinge, CH2 and CH3 regions of mouse IgG2c or rabbit IgG[46].

For the production of phage display libraries, the PCR products were cloned into the pHEN2 phagemid downstream of the pelB-leader peptide and upstream of the chimeric His6x-Myc epitope tag[56]. Phagemids were electroporated into TG1 *E. coli* (Lucigen). Phage particles were precipitated with polyethylene glycol from culture supernatants of *E. coli* transformants infected with a 10-fold excess of M13K07 helper phage (GE Healthcare) (CLEC9A) or hyperphage (Progen) (IgE).

### Production of recombinant antibodies in HEK-6E cells
Five to six days after transient transfection of HEK-6E cells with pCSE2.5 expression vectors, supernatants containing rabbit or mouse hcAbs were harvested and analyzed by reducing SDS-PAGE and

Coomassie staining. Antibodies were purified by affinity chromatography on protein A sepharose columns (GE Healthcare).

### Generation of llama IgG antibody-producing hybridoma cells
Spleen and lymph node cells were fused to Sp2/0 myeloma cells using PEG3350 (Stem Cell Technologies). Fused cells were cultivated on 96-well plates for 10-16 days in the HAT-selection medium. Supernatants of single clones were screened for the production of specific antibodies by ELISA on immobilized AAV particles (see above). The VHH-encoding region was PCR-amplified from cDNA of positive hybridomas and re-expressed as rabbit IgG heavy chain antibodies using the pCSE2.5 expression vector (see above). Supernatants of transfected HEK-6E cells were screened for production of AAV-specific antibodies by ELISA and by immunofluorescence microscopy of HEK-293T cells 48 h after transient co-transfection with AAV expression vectors (rep-cap, pHelper, and an ITR-flanked transgene encoding pscAAVluc). Cells were fixed with 4% paraformaldehyde (PFA) in PBS for 10 min and permeabilized in PBS containing 2% BSA, 3% goat serum, and 0.5% IGEPAL (Sigma-Aldrich) for 1 h. Cells were incubated with HEK-6E supernatants containing rabbit hcAbs for 1 h, washed, and bound antibodies were detected with PE-conjugated anti-rabbit IgG (Dianova). Nuclei were counterstained with DAPI (Sigma) for 10 min. Cells were analyzed with an EVOS Cell Imaging Microscope System.

### Selection of nanobodies on the Berkeley Lights Beacon platform
Recovery of nanobodies from single ASCs was performed automatically on the Beacon platform using the Berkeley Lights Cell Analysis Suite (CAS), essentially as described[44]. ASCs were isolated from the spleen and lymph nodes of AAV8 immunized LamaMice using the CD138+ Plasma Cell Isolation Kit from Miltenyi Biotec following the manufacturer's instructions. Purified B cell samples were imported onto two OptoSelect 11k chips in Berkeley Lights' ASC survival medium that promotes antibody secretion and preserves cell viability. Single-cell penning was then performed using Berkeley Lights' proprietary OEP technology, in which light is used to clone B cells into individual nanoliter-volume chambers (Nanopens). Two antigen-screening mixtures were prepared in medium, containing either 5 μg/ml Dylight 650-conjugated goat anti-llama IgG H + L (ImmunoReagents Inc.) or 5 μg of AlexaFluor647-conjugated AAV8. Primary secreted antibodies were captured on 6−8 μM polystyrene beads (Spherotec) coated with a mouse monoclonal antibody specific to the Fc region of llama IgG2/3 (Southern Biotech). The screening assay was initiated with image capture iterations every five min for a total of ten iterations. After assay completion, the 11k chips containing the ASCs were flushed with a culture medium to remove beads. Plasma cells secreting AAV8-specific antibodies were identified with the software ImageAnalyzer (Berkeley Lights). Following visual hit verification, plasma cells were exported via OEP technology out of the nanopens and flushed one by one into a 96-well export plate containing lysis buffer. RNA purification, cDNA synthesis, and PCR-amplification of the VHH-coding region was performed according to Berkeley Lights proprietary protocols. PCR-amplicons were cloned into an in-house (Genovac) mammalian expression vector containing the hinge, CH2 and CH3 domains of llama IgG2b. Plasmids were transfected into HEK-derivative cells. Antibodies in cell supernatants harvested five days after transfection were screened for specific binding to AAV8 by ELISA.

### Selection of CLEC9A-specific VHH by phage display
Phage libraries were selected in suspension on HEK293T cells transiently transfected with human CLEC9A. Bound phages were eluted by trypsinization and amplified in TG1 *E. coli*. Single colonies were picked, their VHHs were sequenced, and clonal families were determined. Representative members of each family were cloned into the pET22b(+) expression vector for site-specific labelling of monovalent nanobodies or into the pCSE2.5 vector[43] downstream of the IgH-leader

peptide and upstream of the coding regions for the hinge, CH2 and CH3 regions of rabbit IgG[46] to analyze their binding to CLEC9A as bivalent hcAbs.

## Site-specific labelling of VHHs using sortase (eSrtA)

The VHH-encoding region was subcloned into the pET22b(+) expression vector downstream of the pelB signal sequence and upstream of a G4S linker followed by the LPETGG sortagging motif and a hexa-histidine tag. Protein expression in transformed BL21 (DE3) *E. coli* was induced with IPTG during exponential growth. After cultivation of cells at 30 °C for 16 h, periplasmic lysates were generated by osmotic shock and removal of bacterial debris by high-speed centrifugation. Nanobodies were purified by immobilized metal affinity chromatography using Ni-NTA beads[46,56]. Purified nanobodies carrying the sortagging motif were labeled site-specifically with the FITC-conjugated peptide nucleophile GGGCK[FITC]. 100 μg nanobody (20 μM) was modified with 3 M Δ59 eSrtA (0.75 equiv.) and excess nucleophile (25 equiv.) in [50 mM Tris pH 7.5, 150 mM NaCl] supplemented with 10 mM CaCl$_2$ and 5% (v/v) DMSO for 1 h at 37 °C[57]. Unmodified nanobodies were removed by passage over Ni-NTA beads and excess peptide was removed by ultracentrifugation on a 0.5 ml 10 kDa filter (UFC501024, Merck-Millipore). Purity and peptide removal was controlled by SDS-PAGE in-gel fluorescence.

## Nano-differential scanning fluorimetry (nanoDSF)

The thermal and colloidal stability of purified monomeric V$_H$Hs was determined using nano-differential scanning fluorimetry (nanoDSF). To this end, V$_H$Hs at a concentration of 1 mg/ml were loaded into nanoDSF Grade Standard Capillaries (NanoTemper Technologies) and analysed using a Prometheus NT.48 nanoDSF instrument (Nano-Temper Technologies). The V$_H$Hs were subjected to a thermal ramp from 20 to 95 °C in increments of 1 °C/min. Fluorescence at 350 nm and 330 nm the scattering were monitored. The melting temperature $T_m$ was determined using the first derivative of the ratio of fluorescence at 350 and 330 nm. The aggregation onset $Agg_{on}$ was derived from the first derivative of observed scattering.

## Biolayer interferometry (BLI)

The affinity of purified monomeric VHHs was determined using biolayer interferometry (BLI). Using an Octet Red96 (FortéBio/Sartorius) system, biotinylated human IgE (Antibodies Online) or mouse IgE (BP Pharmingen) were immobilized on streptavidin biosensors (Sartorius) and quenched using 10 μg/ml biocytin. Biosensors were incubated first in assay buffer (PBS, 0.1 % BSA, 0.02% Tween-20), then with purified VHHs (in three different concentrations, diluted in assay buffer) and finally with assay buffer. A 1:1 binding model was fitted to the resulting sensorgrams using the Octet BLI Analysis 12.2 software after the deduction of a buffer-only reference. Binding specificities of IgE- and IgG2c-specific VHHs were analyzed using an Octet Red96 or BLItz (ForteBio). Biotinylated VHH were immobilized on streptavidin biosensors (Sartorius), quenched using 10 μg/ml biocytin, and incubated first in assay buffer. Loaded biosensors were then incubated with 20 nM human IgE (Antibodies Online), IgG1 (Southern Biotech), IgG2 (Biolegend), IgG3 (Southern Biotech), IgG4 (Biolegend), IgM (Southern Biotech) or rabbit IgG (Proteintech). Biosensors loaded with VHH374 were incubated with 20 nM mouse IgE (BP Pharmingen 557079), IgG1 (Proteintech 66360-1-Ig), IgG2a (Proteintech 66360-2-Ig), IgG2b (Proteintech 66360-3-Ig), IgG2c (Chromotek 32F6), IgG3 (Helmholtz) and IgM (Southern Biotech 0101-01). Binding responses to the different immunoglobulins were monitored for 120 s (association) followed by 120 s incubation with assay buffer (dissociation).

## SARS-CoV-2 pseudovirus neutralization assay

HEK293T cell supernatants containing luciferase-encoding lentiviral gene ontology (LeGO) vectors[58] pseudotyped with the SARS-CoV-2 spike protein[59] of the Wuhan-Hu-1 (de novo synthesized) or Omicron BA.2 (Addgene 183700) variant were pre-incubated in triplicate samples for 1 h at 37 °C with serial dilutions of HEK cell supernatants containing RBD-specific VHH-rabbit IgG hcAbs before addition to HEK293T cells stably overexpressing the human ACE2 receptor (clone "2G7") seeded in a 96-well cell culture plate (2 × 10$^4$ cells/well). Cells without the addition of pseudovirus were used as a negative control, and cells with the addition of pseudovirus not pre-incubated with RBD-specific VHH-rabbit IgG hcAbs were used as a positive control. After two days of incubation at 37 °C, 5% CO$_2$, and high humidity, luciferin diluted in DPBS was added to each well (0.25 mM final concentration, Biosynth, L-8220, Compton, UK). After 20 min, relative light units (RLU) were analyzed in a Tecan Infinite F200 PRO plate reader (Tecan Lifesciences, CH). The samples' RLU percent were calculated normalized to the negative controls and relative to the positive controls as

$$RLU\,(\%\,of\,untreated) = \frac{RLU_{hcAb} - meanRLU_{no\,vector}}{meanRLU_{untreated}} \times 100$$

## Statistics and reproducibility

Statistical analyses were performed using Microsoft Excel and Graph-Pad Prism 9. The data are presented as mean ± standard deviation (SD) either among the individual mice within a line, or among the individual measurements performed in triplicates. For the pseudovirus neutralization assay, a nonlinear regression curve was generated using the "[Inhibitor] vs. response (three parameters)" model. All experiments for which results from representative experiments are shown, i.e., flow cytometry, immunofluorescence microscopy, ELISA, biolayer interferometry, and SDS-PAGE were performed two to four times.

## Licenses

Schematics of LamaMice in Figs. 3–6 and Sup. Fig. 7 were modified from a picture obtained from Wikimedia under a Creative Common license CC0 1.0 Universal Public Domain Dedication. All other schematics were created using Adobe Illustrator or Affinity Designer software.

## Reporting summary

Further information on research design is available in the Nature Portfolio Reporting Summary linked to this article.

# Data availability

Source data are provided with this paper [https://doi.org/10.6084/m9.figshare.24885546]. The sequences of the BACs were deposited in GenBank [https://www.ncbi.nlm.nih.gov/nuccore/JQ684648] and [https://www.ncbi.nlm.nih.gov/nuccore/JQ684647]. All data needed to evaluate the conclusions in the paper are present in the paper or the Supplementary Materials. Reagents and materials described in this paper are available from the authors upon request, for which a material transfer agreement is to be executed with UKE. Requests should be addressed to F.K.-N. LamaMice are available under a non-exclusive license for academic research. Source data are provided with this paper.

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

## Acknowledgements

This study was funded by the German Research Foundation as part of SFB1328-A10, -A14 and -Z02 (F.H., T.M., B.R., F.K.-N.), and grants BA5893/7, NO310/16 (P.B., F.K.-N.) and TO1013 (N.M.T.), the Federal Ministry of Education and Research as part of COMMUTE (F.K.-N.) and CIM (K.F.H., F.K.-N.), the Werner Otto foundation (J.We., F.K.-N.), the European Research Council grants 679921, 834618 (M.V., C.G.F.), the Dutch Research Council (M.V.), and an Erasmus+ grant (C.M.L.G.). We thank Birte Albrecht and Joanna Schmidt, Institute of Immunology, UKE, and Maria Kuschel, Andrea Nitsche, Nicole Lüder, and Sarah Hewald, Zentrale Versuchstierhaltung, UKE, for excellent technical support. We thank the following scientists for providing reagents: Michel Cogné, Limoges, France, for providing mouse LCR; Neal Copeland, NIH, Maryland, USA, for providing *E. coli* SW106; Thomas Schirrmann, Braunschweig, Germany, for providing pCSE2.5 vector; Yves Durocher NRC, Canada, for providing HEK-6E cells; Klaus Rajewsky, Berlin, Germany, for providing JHT mice.

## Author contributions

T.E., J.We., N.T., I.H.-B., and L.D. cloned and engineered llama BACs and generated LamaMice. T.E., A.Z.S., and N.T. analyzed B cell development in LamaMice. T.E., A.Z.S., T.S., N.T., N.R., W.S., J.H., J.Wo., C.M.L.G., S.W., C.L.-W., M.S., I.B., A.W., Y.H., Le.S., La.S., D.Z., E.S., F.S., A.J.G., A.M.M., and S.M. performed the nanobody discovery campaigns, and produced and characterized recombinant nanobodies and heavy chain antibodies. C.M.L.G., K.R., B.F., P.B., T.M., M.V., C.G.F., K.F.H., H.S., J.F., N.M.T., F.H., B.R., F.K.-N. provided advice and/or acquired grant support. T.E., F.K.-N. supervised this study. T.E., A.Z.S., F.K.-N. wrote the manuscript.

## Funding

## Competing interests

T.E., J.We., S.M., and F.K.-N. are co-inventors on a patent application WO2018/104528 on LamaMice. S.W., C.L.-W., M.S., I.B., A.W., Y.H., Le.S., La.S., D.Z., E.S., K.F.H., H.S., and J.F. work for companies that commercialize products in the field of nanobodies. The remaining authors declare no competing interests.
