## [Peer Review File · Nature Communications]

Generation of nanobodies from transgenic ‘LamaMice’
lacking an endogenous immunoglobulin repertoireREVIEWER COMMENTS

Reviewer #1 (expert in nanobody production):

This paper is a comprehensive description on the generation of a transgenic mice (referred to LaMice) producing heavy-chain only antibodies (HCAbs). After immunising the LaMice, useful VHH (the antigen binding domain of HCAbs) or HCAbs can be retrieved either by hybridoma technology, or phage display selections, single B cell selections or direct cloning into expression vectors. A wide variety of antigens and immunisations strategies were tested. In each case binders were identified that turned out to be target specific (or cross reactive with mutants/variants of the immunogen).

This LaMice seems better than previously generated transgenic mice producing heavy chain only antibodies. The distinction and improvements between this LaMice and the other tg mice is well explained in the discussion part.

The methodology part is highly detailed and is apparently correct in all aspects. The amount of work is not the type that can be reproduced on a rainy Saturday afternoon, but the description is of sufficient quality that it can be reproduced after years of work to generate the engineered genomic BAC construct, the tg mice, the immunisations and the selections of the binders.

Reviewer #2 (expert in animal models and transgenesis):

This manuscript aims to develop a mouse model that contains llama IgH antibodies (nanobodies) in place of the endogenous antibody production system. The authors use a simple approach by using a BAC containing the modified llama IgH locus randomly integrated and then crossed to B cell deficient mice to generate LaMice. This system represents a potential powerful tool to generate and characterize monoclonal nanobodies efficiently without having to use llamas themselves. Many of the experiments demonstrate potential evidence that the LaMice produce nanobodies that have specificity against target antigens. However, there are many details about how the LaMice mouse was generated that are significantly lacking and thus it is difficult to interpret the data presented in the manuscript.

There are essentially no details in the methods for how the animals are housed, cared for, or maintained. Please amend the methods in accordance with the ARRIVE 2.0 guidelines.

Why are balbC mice used for control animals? BALB/c mice are not related to the strains used to make these mice. An additional control needed is to show that the JHT mice or BAC negative litter mate controls do not have B cells/cell surface marker expression as expected. The mice generated have multiple background strains (JHT – B6.129P2) while they BAC was injected into B6xCBA hybrid mice. The LaMice would not be white as depicted in the figures throughout the paper.

Similarly, supplemental figure 2b uses B6J and BALB/c mice as controls which are not relevant. A better control would be the B6xCBA hybrids crossed to the JHT mice.

Supplemental figure 3, an additional control of the JHT mice is needed to show that with these flow conditions no B cells are observed.

The methods discuss founder mice but do not describe how many founders were used to backcross to the JHT mice. How many founders were identified for TE02 and TE03? Each founder mouse represents a random integration of the BAC that is distinct from other potential founder mice in the same injection. Where the founders interbred? Was each line maintained as hemizygous or homozygous for the BAC?

What are the integration sites for the BACs? Are the BACs integrated in the correct structure? How many copies of the BAC are integrated for each founder.

In the breeding scheme described, the BAC founder mice were backcrossed to the JHT mice, was this done multiple times? A single cross would lead to heterozygous JHT mice that are BAC positive but would not be deficient for B cells.

Reviewer #3 (expert in nanobody production):

In the manuscript “Nanobody discovery from immunized VHH-transgenic LaMice by classic hybridoma technology, single B cell screening, direct cloning, and phage display” the authors described obtaining transgenic mice expressing only llama’s single domain antibodies (sdAb). Obtaining transgenic mice is well detailed and scientifically based. The presence of functional B cells displaying llama heavy chain Immunoglobulins is well established including the presence of somatic hypermutations at the genetic level showing that the mouse genetic machinery is active on llama genes. They further demonstrate that the immunized mice can be a source of specific monoclonal antibodies by using different well-established techniques. Several types of antigens were used for immunization, demonstrating that these LaMice could be an alternative to camels for obtaining single domain antibodies.

However, several points need to be clarified.

- No indication is given on the isotype of sdAb (IgM, IgG) present in the LaMice before and after immunization. Please specify.
- Mice were immunized with recombinant AAV and clones were obtained either by cell fusion or by beacon technology. It appears, but it’s not clear, that the antibodies are different using the two technologies. If so, is it related to the use of different LaMice strains? What is the rationale for the use of one strain instead of the other?
- 22/28 clones use the same VHH5 gene. Is it due to the fact that the VHH5 gene is proximal to the D and J genes ? the most proximal V gene is a VH gene and was used once (supp table 1). The VH genes are not found in sdAb except VH gene from the VH4 family found in both types of Igs. Please clarify and discuss.
- The main concern, however, lies in the fact that the nanobodies are only presented as sdAb. This presentation does not reflect the biochemical properties of nanobodies mainly the affinity, because sdAb have generally better avidity for the antigen. The interest of the LaMice is to provide high quality nanobodies but in the present manuscript no data are given about the nanobodies per se. These data are missing and need to be added.

Other specific comments

- Due to the importance of data presented in Supp table 1, it is important to include them in the full text.
- No indications are given about the parental mouse line used to establish the LaMice strains. Please specify

- In the discussion paragraph, the authors consider the small size of the B cell repertoire in mice and suggest the use of a larger numbers of LaMice. However, one mouse line shares the same genetic background, and the repertoire could be almost identical. Please comment

Point by point response to REVIEWER COMMENTS

Reviewer #1 (expert in nanobody production):

This paper is a comprehensive description on the generation of a transgenic mice (referred to LaMice) producing heavy-chain only antibodies (HCAbs). After immunising the LaMice, useful VHH (the antigen binding domain of HCAbs) or HCAbs can be retrieved either by hybridoma technology, or phage display selections, single B cell selections or direct cloning into expression vectors. A wide variety of antigens and immunisations strategies were tested. In each case binders were identified that turned out to be target specific (or cross reactive with mutants/variants of the immunogen).

This LaMice seems better than previously generated transgenic mice producing heavy chain only antibodies. The distinction and improvements between this LaMice and the other tg mice is well explained in the discussion part.

The methodology part is highly detailed and is apparently correct in all aspects. The amount of work is not the type that can be reproduced on a rainy Saturday afternoon, but the description of of sufficient quality that it can be reproduced after years of work to generate the engineered genomic BAC construct, the tg mice, the immunisations and the selections of the binders.

We thank reviewer #1 for her/his wink-of-the-eye appreciation of the considerable time and work that we put into developing and evaluating LaMice.

Reviewer #2 (expert in animal models and transgenesis):

This manuscript aims to develop a mouse model that contains llama IgH antibodies (nanobodies) in place of the endogenous antibody production system. The authors use a simple approach by using a BAC containing the modified llama IgH locus randomly integrated and then crossed to B cell deficient mice to generate LaMice. This system represents a potential powerful tool to generate and characterize monoclonal nanobodies efficiently without having to use llamas themselves. Many of the experiments demonstrate potential evidence that the LaMice produce nanobodies that have specificity against target antigens. However, there are many details about how the LaMice mouse was generated that are significantly lacking and thus it is difficult to interpret the data presented in the manuscript.

There are essentially no details in the methods for how the animals are housed, cared for, or maintained. Please amend the methods in accordance with the ARRIVE 2.0 guidelines.

1. As suggested, we added the following § to the Methods section:

Animal experiments

Animal experiments were performed according to national and institutional animal care and ethical guidelines and were approved by the Veterinarian Agency of Hamburg and the local animal care committee (registration number A029/2019). All mice were maintained in a specific-pathogen-free facility at temperatures of 21–24°C with 40-70% humidity on a 12 h light/12 h dark cycle and provided with food and water ad libitum. When applicable, animal

experiments were conducted in accordance with the ARRIVE 2.0 guidelines. The number of animals used for experiments is specified in the respective figure legends. As these studies were largely exploratory in character, no sample size was calculated a priori. No specific inclusion or exclusion criteria applied. No animals and no data points were excluded from the analyses. Animals were randomized if feasible (e.g. for the comparison of non-immunized with immunized mice from the same genetic background). Investigators were blinded for biological analyses (e.g. FACS stainings) and data analyses.

Why are balbC mice used for control animals? BALB/c mice are not related to the strains used to make these mice. An additional control needed is to show that the JHT mice or BAC negative litter mate controls do not have B cells/cell surface marker expression as expected. The mice generated have multiple background strains (JHT – B6.129P2) while they BAC was injected into B6xCBA hybrid mice. The LaMice would not be white as depicted in the figures throughout the paper.

2. BALB/c mice are commonly used for generating monoclonal antibodies (PMID: 35820791). BALB/c mice (H2^d) have two functional MHCII loci (IA^d and IE^d), whereas B6 mice (H2^b) carry a null allele of the IE locus (PMID: 6296871). We reasoned additional MHCII loci of the H2d/d or H2d/b haplotypes would allow B cells of LaMice to present a larger pool of peptides and therefore receive more effective T-cell help. We also transferred the H2^b locus to JHT mice by crossing to BALBc. We added this information to the Methods.

LaMice, indeed come in various colors. We modified the representation of LaMice in the figures.

Greenfield EA. Immunizing Animals. Cold Spring Harb Protoc. 2022 Jul 12;2022(7):Pdb.top100180. doi: 10.1101/pdb.top100180. PMID: 35820791.

Mathis DJ, Benoist C, Williams VE 2nd, Kanter M, McDevitt HO. Several mechanisms can account for defective E alpha gene expression in different mouse haplotypes. Proc Natl Acad Sci U S A. 1983 Jan;80(1):273-7. doi: 10.1073/pnas.80.1.273. PMID: 6296871.

Similarly, supplemental figure 2b uses B6J and BALB/c mice as controls which are not relevant. A better control would be the B6xCBA hybrids crossed to the JHT mice.

3. As suggested, we expanded Fig. 2b to include CBA, BALBc, B6J, and JHT mice as controls. As indicated in response 2, we backcrossed LaMice from the F1 CBA/B6 background to BALBc mice in order to transfer the intact BALBc MHC locus (H2^d) to LaMice. All immunized LaMice were homozygous for the JHT mutation of the IgH locus, homo or heterozygous at the MHC locus (H2d/d or H2d/b), homo- or hemizygous for the BAC, and carry different combinations of other genetic loci from B6, CBA, or BALBc mice.

Supplemental figure 3, an additional control of the JHT mice is needed to show that with these flow conditions no B cells are observed.

4. As suggested, we added controls of JHT mice to Sup. Fig. 3.

The methods discuss founder mice but do not describe how many founders were used to backcross to the JHT mice. How many founders were identified for TE02 and TE03? Each founder mouse represents a random integration of the BAC that is distinct from other

potential founder mice in the same injection. Where the founders interbred? Was each line maintained as hemizygous or homozygous for the BAC?

5. Two founders were identified for TE02 and four founders for TE03. The founders were not interbred. The number of circulating B cells were monitored by flow cytometry of a blood sample taken at 6-8 weeks of age (Supplementary Fig. 2b). For each line, we chose a founder whose offspring consistently showed a high number of circulating B cells. In order to transfer the intact BALB/c MHC locus (H2^d) to LaMice, we backcrossed TE02, TE03, and JHT mice to BALB/c mice. Mice that are homozygous for the BAC are vital and fertile, indicating that the site of integration did not disrupt any essential gene. TE02 and TE03 LaMice are now maintained by interbreeding as hemizygous and homozygous for the BAC. Every 6-10 generations, the lines are backcrossed to JHT mice. We added this information to the Methods.

What are the integration sites for the BACs? Are the BACs integrated in the correct structure? How many copies of the BAC are integrated for each founder.

6. Due to budget limitations, we did not determine the integration sites of the BACs. By breeding animals that were homozygous for both, the JHT locus and the BAC, we verified that the BAC had not disrupted the function of an essential gene.

We requested quotes from several CROs to perform these analyses. The estimated costs and time are painfully high. Since the integrated BACs are functional, information about their integration site and copy number will not affect any of the results or conclusions presented in the paper. Therefore, we kindly request publication of our paper without this information.

In the breeding scheme described, the BAC founder mice were backcrossed to the JHT mice, was this done multiple times? A single cross would lead to heterozygous JHT mice that are BAC positive but would not be deficient for B cells.

7. Yes, the mice were backcrossed multiple times to JHT mice and are maintained as homozygous for the JHT mutation. Further, as indicated in responses 2 and 5 above, we also backcrossed JHT mice with BALBc mice in order to transfer and maintain the BALBc MHC locus (H2^b) when backcrossing LaMice with JHT mice.

Reviewer #3 (expert in nanobody production):

In the manuscript “Nanobody discovery from immunized VHH-transgenic LaMice by classic hybridoma technology, single B cell screening, direct cloning, and phage display” the authors described obtaining transgenic mice expressing only llama’s single domain antibodies (sdAb). Obtaining transgenic mice is well detailed and scientifically based. The presence of functional B cells displaying llama heavy chain Immunoglobulins is well established including the presence of somatic hypermutations at the genetic level showing that the mouse genetic machinery is active on llama genes. They further demonstrate that the immunized mice can be a source of specific monoclonal antibodies by using different well-established techniques. Several types of antigens were used for immunization, demonstrating that these LaMice could be an alternative to camels for obtaining single domain antibodies.

However, several points need to be clarified.

- No indication is given on the isotype of sdAb (IgM, IgG) present in the LaMice before and after immunization. Please specify.

1. Using llama IgG- and IgM-isotype-specific primers for RT-PCR analyses, we readily detect IgG and IgM transcripts in both, naïve and immunized LaMice. Representative results are shown in the new panel d of Supplementary Fig. 2.

- Mice were immunized with recombinant AAV and clones were obtained either by cell fusion or by beacon technology. It appears, but it's not clear, that the antibodies are different using the two technologies. If so, is it related to the use of different LaMice strains? What is the rationale for the use of one strain instead of the other?

2. We do not observe any fundamental differences in nanobodies selected with the two technologies or in nanobodies selected from the two strains of LaMice. There is no particular rationale for the use of one strain or the other.

- 22/28 clones use the same VHH5 gene. Is it due to the fact that the VHH5 gene is proximal to the D and J genes? The most proximal V gene is a VH gene and was used once (supp table 1). The VH genes are not found in sdAb except VH gene from the VH4 family found in both types of Igs. Please clarify and discuss.

3. Yes, the preferential usage of the VHH5 gene could, conceivably, be related to its proximity to the D and J genes. As in case of llamas (PMID: 18641337), the most proximal VH gene (designated VH3-1) is rarely used in hcAbs in LaMice. However, the AAV8-specific VH-nanobody (8-A110-3) that we selected from immunized LaMice, shows robust stability and solubility. This is similar to VH-nanobodies of known 3D structure that have been co-crystallized with their cognate antigen (e.g. pdb code 6glw:C, 8h3x:A). As suggested, we added a corresponding § to the Discussion to clarify and discuss this point.

Achour I, Cavelier P, Tichit M, Bouchier C, Lafaye P, Rougeon F. Tetrameric and homodimeric camelid IgGs originate from the same IgH locus. *J Immunol.* 2008 Aug 1;181(3):2001-9. doi: 10.4049/jimmunol.181.3.2001. PMID: 18641337.

- The main concern, however, lies in the fact that the nanobodies are only presented as sdAb. This presentation does not reflect the biochemical properties of nanobodies mainly the affinity, because sdAb have generally better avidity for the antigen. The interest of the LaMice is to provide high quality nanobodies but in the present manuscript no data are given about the nanobodies per se. These data are missing and need to be added.

4. We apologize for not having made this sufficiently clear. The results presented in the paper do include data on monovalent nanobodies (Fig. 5c, Fig. 6c, Supplementary Fig. 7c, for IgE-, CLEC9a, and IgG2c-specific nanobodies). We added small schematics of monomeric nanobodies and bivalent hcAbs to all figures to clarify this.

Further, as suggested, we added more data on the biochemical properties of nanobodies obtained from LaMice and alpacas using nano-differential scanning fluorimetry to determine their thermal stability and aggregation behaviour and biolayer interferometry (BLI) to estimate their affinity. The results (presented in new Supplementary Fig. 7) show that nanobodies from LaMice and nanobodies from alpacas display a similar range of biochemical properties. Indeed, two IgE-specific nanobodies obtained from LaMice were chosen for

commercialization by Chromotek on the basis of their excellent biochemical properties. The results underscore the high quality of nanobodies obtained from LaMice.

A key advantage of nanobodies over the paired VH and VL domains of conventional antibodies is that they can be readily converted into monovalent, bivalent, and multivalent formats. Nanobodies derived from LaMice can be converted just as easily as nanobodies from alpacas into any such format. We added two sentences to the Discussion and a schematic diagram (new Supplementary Fig. 10) to discuss and clarify this point.

Other specific comments

- Due to the importance of data presented in Supp table 1, it is important to include them in the full text.

5. As suggested, we moved Supplementary Table I to the full text.

- No indications are given about the parental mouse line used to establish the LaMice strains. Please specify

6. The BACs were initially injected into F1 hybrid oocytes (B6xCBA). Founders were backcrossed to JHT mice (for the inactivated endogenous IgH locus) and to BALB/c mice (for the functional MHCII locus). We specified this in the Methods section.

- In the discussion paragraph, the authors consider the small size of the B cell repertoire in mice and suggest the use of a larger numbers of LaMice. However, one mouse line shares the same genetic background, and the repertoire could be almost identical. Please comment

7. Nanobodies obtained from the same mouse often show multiple variants of a single clone (subclonotypes), i.e. intraclonal diversification due to somatic hypermutation (PMID: 37391484). Random addition and deletion of nucleotides at the V-D and D-J junctions during B cell development in the bone marrow ensures that even identical twins develop their own, unique B cell repertoires. Clones with quasi-identical “stereotyped” sequences from two different mice (metaclonotypes) have been documented in the literature, e.g. upon immunization with simple hapten antigens (PMID: 26194752). To date, we have not observed any metaclonotypes amongst dozens of nanobodies selected from distinct LaMice - with the possible exception of the RBD-specific nanobody families 2 and 3 which carry an unusually short and similar CDR3 sequence of only three amino acids (Table I).

Sofou E, Vlachonikola E, Zaragoza-Infante L, Brüggemann M, Darzentas N, Groenen PJTA, Hummel M, Macintyre EA, Psomopoulos F, Davi F, Langerak AW, Stamatopoulos K. Clonotype definitions for immunogenetic studies: proposals from the EuroClonality NGS Working Group. *Leukemia*. 2023 Aug;37(8):1750-1752. doi: 10.1038/s41375-023-01952-7. Epub 2023 Jun 30. PMID: 37391484.

Henry Dunand CJ, Wilson PC. Restricted, canonical, stereotyped and convergent immunoglobulin responses. *Philos Trans R Soc Lond B Biol Sci*. 2015 Sep 5;370(1676):20140238. doi: 10.1098/rstb.2014.0238. PMID: 26194752

REVIEWER COMMENTS

Reviewer #2 (expert in animal models and transgenesis):

Thank you to the authors for the changes they have made in the revised manuscript. In the rebuttal, they state the following in response 5:

"Mice that are homozygous for the BAC are vital and fertile, indicating that the site of integration did not disrupt any essential gene."

However, this does not mean there are not adverse effects of the BAC integration site(s). Goodwin et al Genome Research PMID 30659012 highlighted that after identification of random Transgene insertion sites for either short transgenes or BACs in viable lines that are commercially available, the integration event can have significant ramifications on the overall genotype and phenotype of the animal.

Additionally, when BACs integrate, they often undergo fragmentation and local chromothripsis with host DNA being intermingled with BAC DNA and having impacts on the overall structure of the BAC (DuBose et al Nucleic Acids Research, 2013, PMID 23314155). Supplemental figure 2a does use PCR to characterize some sections of the BAC, although it is unclear whether this figure represents founders for the TE02 or TE03 BACs. Additionally, the PCR products represent a very small fraction of the entire BAC and no sequencing of the PCR products was done to confirm that the structure of the sections is as expected. For these BACs, structural rearrangements could have impacts on the way the VhH usage or the types of antibodies that are generated.

While these facts may not directly impact the findings in this study, in the discussion the authors state the following:

"For example, LaMice can be crossed with target-deficient mouse lines or target genes could be deleted in LaMice by CRISPR/Cas or similar technologies. Provided that the physiological antibody response is not compromised by the defective target gene, immunization of such target-deficient LaMice with the human gene product is expected to yield antibodies also against conserved epitopes"

and

"LaMice thus combine powerful technologies that facilitate nanobody discovery for broad applications in biotechnology and medicine."

If these mice are going to be broadly used as the authors wish, the issues described above are critical to ensure the application of these mice is not confounded by issues associated with integration of the BAC. Further characterization of the BAC structure and integration site are needed.

Reviewer #3 (expert in nanobody production):

The authors clearly answered all questions asked.

This article do deserves to be published in Nature Communications

REVIEWER COMMENTS

Reviewer #2 (expert in animal models and transgenesis):

Thank you to the authors for the changes they have made in the revised manuscript. In the rebuttal, they state the following in response 5:

"Mice that are homozygous for the BAC are vital and fertile, indicating that the site of integration did not disrupt any essential gene."

However, this does not mean there are not adverse effects of the BAC integration site(s). Goodwin et al Genome Research PMID 30659012 highlighted that after identification of random Transgene insertion sites for either short transgenes or BACs in viable lines that are commercially available, the integration event can have significant ramifications on the overall genotype and phenotype of the animal.

Additionally, when BACs integrate, they often undergo fragmentation and local chromothripsis with host DNA being intermingled with BAC DNA and having impacts on the overall structure of the BAC (DuBose et al Nucleic Acids Research, 2013, PMID 23314155). Supplemental figure 2a does use PCR to characterize some sections of the BAC, although it is unclear whether this figure represents founders for the TE02 or TE03 BACs. Additionally, the PCR products represent a very small fraction of the entire BAC and no sequencing of the PCR products was done to confirm that the structure of the sections is as expected. For these BACs, structural rearrangements could have impacts on the way the VhH usage or the types of antibodies that are generated.

While these facts may not directly impact the findings in this study, in the discussion the authors state the following:

"For example, LaMice can be crossed with target-deficient mouse lines or target genes could be deleted in LaMice by CRISPR/Cas or similar technologies. Provided that the physiological antibody response is not compromised by the defective target gene, immunization of such target-deficient LaMice with the human gene product is expected to yield antibodies also against conserved epitopes"

and

"LaMice thus combine powerful technologies that facilitate nanobody discovery for broad applications in biotechnology and medicine."

If these mice are going to be broadly used as the authors wish, the issues described above are critical to ensure the application of these mice is not confounded by issues associated with integration of the BAC. Further characterization of the BAC structure and integration site are needed.

Response:

As suggested, we have now determined the BAC structure and integration sites. The analyses were performed by Cergentis, Utrecht, NL (de Vree et al. Nat Biotechnol. 2014, PMID 25129690). The results show that BAC TE02 integrated in the *Abca5* gene on Chromosome 11. A 24 kb genomic DNA fragment (chr11:110,311,269-110,335,224) encompassing exons 2-9 of *Abca5* was deleted at the integration site. BAC TE03 integrated downstream of the *Tmem232* gene on Chromosome 17. A 17 kb genomic DNA fragment (chr17:65,372,523- 65,645,975) encompassing an intergenic region was duplicated at the integration site. In both cases, DNA sequencing verified the presence of the entire ~140kb transgene.

de Vree PJ, et al. 2014. Targeted sequencing by proximity ligation for comprehensive variant detection and local haplotyping. Nat Biotechnol. 32:1019-25. PMID: 25129690.

We have added this information in the Materials and Methods section.

After reaching adulthood, *Abca5*^{-/-} mice reportedly develop cardiomyopathy and symptoms of lysosomal storage diseases (Kubo et al. Mol Cell Biol 2005, PMID 15870284). Heterozygous *Abca5*^{+/-} mice do not display any apparent abnormalities. *Tmem232*^{-/-} mice are viable and display no apparent abnormalities (He et al. Cells 2023, PMID: 37371084.) However, inactivation of *Tmem232* induced male sterility but did not influence female fertility.

Kubo Y, et al. 2005. ABCA5 resides in lysosomes, and ABCA5 knockout mice develop lysosomal disease-like symptoms. Mol Cell Biol. 25:4138-49. PMID: 15870284.

He X, et al. 2023 Deficiency of the Tmem232 Gene Causes Male Infertility with Morphological Abnormalities of the Sperm Flagellum in Mice. Cells. 12:1614. PMID: 37371084.

In light of these findings, we added the following sentence to the penultimate paragraph of the Discussion:

"Considering that homozygosity at the BAC integration site may impair health, it is prudent to use heterozygous LaMice for nanobody discovery campaigns."

Reviewer #3 (expert in nanobody production):

The authors clearly answered all questions asked.

This article do deserves to be published in Nature Communications

REVIEWERS' COMMENTS

Reviewer #2 (Remarks to the Author):

Thank you got determining the integration sites of the BACs. I recommend accepting this manuscript.